# The Importance of the Slaughterhouse in Surveilling Animal and Public Health: A Systematic Review

**DOI:** 10.3390/vetsci10020167

**Published:** 2023-02-20

**Authors:** Juan García-Díez, Sónia Saraiva, Dina Moura, Luca Grispoldi, Beniamino Terzo Cenci-Goga, Cristina Saraiva

**Affiliations:** 1Veterinary and Animal Research Centre (CECAV), University of Trás-os-Montes e Alto Douro, Quinta de Prados, 5000-801 Vila Real, Portugal; 2Associate Laboratory for Animal and Veterinary Sciences (AL4AnimalS), Portugal; 3Divisão de Intervenção de Alimentação e Veterinária de Vila Real e Douro Sul, Direção de Serviços de Alimentação e Veterinária da Região Norte, Direção Geral de Alimentação e Veterinária, Lugar de Codessais, 5000-567 Vila Real, Portugal; 4Dipartimento di Medicina Veterinaria, Università degli Studi di Perugia, 06126 Perugia, Italy; 5Faculty of Veterinary Science, Department of Paraclinical Sciences, University of Pretoria, Onderstepoort 0110, South Africa

**Keywords:** cattle, swine, sheep, goat, food safety, zoonoses, animal welfare, antimicrobial resistance, foodborne, one health

## Abstract

**Simple Summary:**

Food safety and quality are the main demands of consumers. Moreover, clear, truthful, and direct information about food, based on science, is essential to build trust among consumers and advance food safety. Traditionally, the role of the slaughterhouse is guaranteeing the safety of meat from the perspective of animal pathology and disease. However, it can be used for monitoring other aspects that influence not only the animal health (One Health), but also the food safety and public health. The present review discusses the role of the slaughterhouse to guarantee the food safety and monitor aspects, such as animal welfare, antimicrobial resistance, or prevalence of foodborne and zoonotic diseases.

**Abstract:**

From the point of public health, the objective of the slaughterhouse is to guarantee the safety of meat in which meat inspection represent an essential tool to control animal diseases and guarantee the public health. The slaughterhouse can be used as surveillance center for livestock diseases. However, other aspects related with animal and human health, such as epidemiology and disease control in primary production, control of animal welfare on the farm, surveillance of zoonotic agents responsible for food poisoning, as well as surveillance and control of antimicrobial resistance, can be monitored. These controls should not be seen as a last defensive barrier but rather as a complement to the controls carried out on the farm. Regarding the control of diseases in livestock, scientific research is scarce and outdated, not taking advantage of the potential for disease control. Animal welfare in primary production and during transport can be monitored throughout ante-mortem and post-mortem inspection at the slaughterhouse, providing valuable individual data on animal welfare. Surveillance and research regarding antimicrobial resistance (AMR) at slaughterhouses is scarce, mainly in cattle, sheep, and goats. However, most of the zoonotic pathogens are sensitive to the antibiotics studied. Moreover, the prevalence at the slaughterhouse of zoonotic and foodborne agents seems to be low, but a lack of harmonization in terms of control and communication may lead to underestimate its real prevalence.

## 1. Introduction

Food safety is an issue of increasing interest and concern worldwide. Public health issues related to food safety can become a risk to consumers at any stage of the food chain. Thus, the World Organization of Animal Health (WOAH) recognizes food safety in livestock production as one of its most pressing priorities [1]. The importance of animal health control has several objectives, such as (i) guaranteeing the optimal veterinary status of animals and promoting their production in terms of animal welfare, (ii) maintaining livestock trade at the regional, national, and international level, (iii) supporting the production of high-quality products of animal origin based on the European food safety philosophy “from farm to fork”, (iv) guaranteeing the public health by preventing zoonoses and foodborne diseases, and (v) promoting sustainability and competitiveness of the livestock sector in an increasingly global market, improving environmental conservation [2,3,4,5,6,7,8].

The guarantee of food safety of food of animal origin is achieved through the implementation of several controls by veterinary authorities throughout the food chain, including primary production, slaughter, food processing industry (i.e., dairy, fishery or meat industry among others), and retail establishments as defined by law [9,10,11,12,13].

In the slaughterhouse, the objective of veterinary inspection services is to control the entire process from the entry of the animals to obtaining the meat. The veterinary meat inspector, after ante-mortem and post-mortem inspection declares it fit for human consumption if it does not present any risk. Moreover, to guarantee the hygiene and healthiness of meat, it is necessary not only to achieve a high level of animal health standards in primary production animals destined for human consumption, but also to implement hygienic and preventive measures in slaughterhouses.

As defined by law, a slaughterhouse is an establishment used for slaughtering and dressing animals whose meat is intended for human consumption [10]. According to the WOAH, slaughterhouses play a key role in the epidemiological surveillance of zoonoses and all other animal diseases, where the official veterinary inspector (OVI) is essential [14]. Therefore, the control activities can be summarized in four steps: (1) analysis of livestock documentation (e.g., food chain information), (2) ante-mortem inspection, (3) post- mortem inspection, and (4) feedback information. *Ante-mortem* inspection is defined as any procedure or test carried out by an OVI on live animals at slaughterhouse in order to issue an opinion on their safety and suitability for human consumption [15]. At this stage, the OVI also verifies all the documents related to the animals, such as official movement forms, animal identification cards, food chain information form, or animal identification (e.g., verification of ear tags and/or electronic identification compliance) among others. 

During the slaughter process, the OVI observes the different slaughter operations are carried out under hygienic conditions in accordance with good manufacture practices [10] Then, post-mortem inspection aims to examine the carcass and viscera, the control of which significantly reduces the spread of diseases and interrupts transmission cycles. All these activities contribute to preventing the spread of epizootics animal diseases and preventing the transmission of zoonotic diseases to humans [16].

However, the classical inspection methodology, which involves incisions and palpation of various organs, has been questioned due to some limitations, such as the impossibility of detecting microbiological or chemical hazards or due to the high rate of animals slaughtered to make the abattoir operations profitable, with special relevance in pig and poultry slaughterhouses. Because pig and poultry farms are based on all-in/all out schemes (the complete emptying of animals from premises, before cleaning and disinfection without the animals inside before introducing a new batch), a risk-based inspection methodology has been proposed that includes the analysis of food chain information forms to determine the risk level of the herd and, consequently, adapt the inspection pressure depending on the hygiene level of the plant, type of animal species slaughtered, or specific tests to detect potential hazards resulting from a risk analysis [17]. Based on this risk assessment, it has been suggested that visual-only meat inspection can safely replace traditional inspection in pigs [18,19], poultry [20], and lambs [21]. 

The visual-only inspection differs, from the classical meat inspection, in the absence of incisions or palpations to avoid cross contamination of the carcasses. Its implementation is based on the impossibility to detect current hazards with high foodborne risk, such as *Samonella* spp, *Campylobacter*, or *Yersinia* spp. [21]. A specific policy in the European Union (EU) [13] indicates that post-mortem inspection in domestic pigs should only be visual. If the OVI suspects the presence of an injury that could put animal health, public health, or animal welfare at risk, they can perform palpations and cuts of swine carcass. Thus, classical meat inspection is still necessary to guarantee animal health and food safety as already demonstrated in the early detection of an outbreak of foot-and-mouth disease in pigs at a slaughterhouse in England [22]. In addition, other lesions in pigs, such as melanosis in mammary parenchyma, osteomyelitis, or caseous lymphadenitis, are detectable by classical meat inspection [23]. 

Since red meat (cattle, sheep, goats) production schemes are variable, the conversion of the traditional meat inspection into a visual-only meat inspection may pose a public health risk with respect to some diseases such as cysticercosis, bovine tuberculosis or fasciolosis, among others [24]. Moreover, undetected cases of endocarditis and embolic pneumonia have been reported in swine [25]. In addition, the substitution of classic meat inspection by visual-only inspection can only be applied in controlled farms (e.g., indoor farms) that are facing the new purchasing trends of consumers who prefer fresh meat and/or meat products from animals raised under adequate welfare conditions (i.e., extensive and/or free-range management) [26,27]. It has been also observed that remote post-mortem veterinary meat inspection through an augmented-reality live-stream in pigs displayed similar results concerning condemnation rates when compared with classical meat inspection. However, confidence about carcass condemnations in doubtful lesions was higher in OVI in slaughterhouse [28]. Another constrain of visual-only meat inspection is related to the insufficient information provided in the compulsory food chain information form regarding animal health and food safety for swine and poultry [29,30,31]. Thus, classical meat inspection has allowed the detection of epidemiological diseases, such as classical swine fever or foot-and-mouth disease, as mentioned above, after the failure of other epidemiological surveillance systems [32].

Disposal by-products of slaughterhouses may impact animal and public health. In the EU, all by-products are destroyed by incineration. Different operations, such as classification, traceability, transportation, and destruction of by-products, are defined by law [13,33] and controlled by national veterinary agencies. Moreover, OVI is responsible for the monitorization of correct classification of by-products and its traceability at slaughterhouse. In other countries (mainly in developing countries) burial and landfilling is a common practice [34]. This practice may imply certainly risk for animal and public health specially in scavenging wildlife health and free-roaming scavengers, which in turn, has implications for human and livestock health [35]. Further, land filling (i.e., solid waste exposure) directly impacts the public health in terms of air, water, and soil pollution [36]. Slaughterhouses, as enterprises, contribute to the environmental impact of waste they produce. Thus, the development of methodologies to minimize this impact, such as transformation into high-added valued food products or energy production, has been studied [36,37,38]. 

Regardless of the transcendental aspects of animal and zoonotic disease control, food safety and public health, the slaughterhouse provides an important source of information for livestock health. Records of causes of condemnations in those countries with adequate meat inspection are essential for epidemiological disease surveillance and welfare management. The existence of databases on diseases and pathological processes in slaughterhouses, updated in real time, together with an appropriate livestock traceability database, constitutes the starting point for the development and implementation of control or eradication programs for livestock diseases with high economic and social impact [39,40,41,42,43]. Furthermore, condemnation data in slaughterhouses are valuable for farmers to improve animal health schemes and on-farm welfare conditions [44]. Thus, the present review discusses the importance of the slaughterhouse as a surveillance center for cattle, small ruminant and swine diseases, zoonotic and foodborne pathogens, and as a surveillance center for antimicrobial resistance and animal welfare.

## 2. Surveillance of Livestock Diseases

### 2.1. Surveillance of Cattle Carcass and Organs Condemnation 

To assess the health status of cattle in a specific region, studying the prevalence of condemnation of whole and/or partial carcasses (recorded at slaughterhouses) may be an appropriate approach. The available data on carcass condemnation of cattle are scarce and usually out of date (Table 1). The condemnation rate of cattle carcasses differed among studies, ranging from 0.1% to 1% (average 0.45%), although some studies reported carcass condemnation rates of around 5% [45]. These results must be carefully evaluated due to differences between the prevalence of diseases, the characteristics of the slaughterhouse, the data source, the study period, or the meat inspection criteria, among others [42,46]. It is important to remark that since most reports are based on observational studies, seasonal variations can also influence the condemnation rate. Moreover, worldwide differences about policy related to meat inspection may contribute to this variation. Despite the fact that countries, such as the United States, Canada, Australia, or some countries of the European Union, have national databases on the total cattle condemnation and their cause, only some take advantage of the potential of the information for research purposes. In the present review, scarce research studies on the causes of cattle carcass condemnation were encountered during the literature review. Knowledge of the condemnation rate not only allows us to verify the health status of cattle, but also allow us to develop predictive models on condemnation from which preventive programs can be elaborated [47,48,49]. Regarding causes of carcass condemnation, several conditions have been described as presented in Table 2. According to reviewed data, bovine tuberculosis, among all the causes, displayed the highest rate of carcass condemnation for some authors [43,50]. Other causes of cattle carcass condemnation in more than 30% of cases were malignant lymphoma, multiple abscess, septicemia, bruising, or pneumonia [39,51]. Although these results are difficult to interpret, the high rates of condemnation related to bovine tuberculosis and malignant lymphoma could be associated with the absence of eradication programs.

Furthermore, high condemnations rates are associated with the presence of lesions in different organs. However, other notifiable diseases (e.g., brucellosis), as they do not have pathognomonic lesions, have not been referenced in terms of their prevalence. The condemnation rate above 30% could be explained by the geographical location of the studies and cattle management, where extensive farming predominates. In these regions, the absence of preventive programs (i.e., vaccination and deworming schemes), eradication programs, veterinary support, and biosecurity measures, makes cattle more vulnerable to diseases, such as bovine respiratory syndrome or bovine paratuberculosis. The presence of bruising on the carcass has been described as a welfare indicator both for pre-slaughter and abattoir operations [63]. Although bruises represent a negative economic impact related to partial condemnations, total carcass condemnation is really scarce [51].

**Table 2 vetsci-10-00167-t002:** Main causes of carcass condemnation in cattle.

Cause of Condemnation	Rate (%)	Country	Reference
Bovine tuberculosis	44.00	Tanzania	[57]
	0.60	Tanzania	[58]
	85.00	Ethiopia	[59]
	1.32	Turkey	[40]
Bruising	50.00	Namibia	[51]
*Cysticercus bovis*	16.00	Tanzania	[57]
	5.10	Tanzania	[58]
	2.70	Zambia	[50]
Emaciation/Cachexia	12.08	Italy	[54]
	28.00	Tanzania	[57]
	3.30	Tanzania	[58]
Jaundice	1.20	Tanzania	[58]
Malignant lymphoma	39.60	United States	[47]
	22.30	United States	[62]
Multiple abcessess	2.00	Tanzania	[57]
	0.50	Tanzania	[58]
	33.30	Namibia	[51]
Oedema	11.42	Ethiopia	[59]
Peritonitis	20.13	Italy	[54]
	7.85	United States	[62]
Pneumonia	37.60	United States	[47]
	10.84	United States	[62]
Septicaemia	37.60	United States	[47]
	20.81	Italy	[54]
	9.00	Tanzania	[57]
	10.39	United States	[62]
Tumours	2.85	Ethiopia	[59]
	9.15 *	United States	[62]

* Rate refers to epithelioma.

Carcasses with a high number of bruises are condemned for their unsightly appearance by OVI but they do not represent a microbiological risk for public health [64]. Although the carcass is condemned for bruising (e.g., unsightly appearance), no fine is applied to the owner. It has been described that the presence of bruises can favor the appearance of alterations, such as dark-firm-dry meat [65]. 

Based in the literature reviewed, information about causes and prevalence of ante-mortem condemnations is scarce. Another study [62] reported some causes in cattle, such as dead animals in pens (64%), moribund cattle (32.4%), and epithelioma (1.8%). As seen before, the total carcass condemnation of cattle represents an important impact for farmers [66]. 

In addition, identifying potential risk factors on the farm may help them improve some measures to decrease condemnation rates. Some factors, such as compliance with a quality meat scheme [48], year, season, price, intended production (meat or dairy), age, or farm audit [53], may influence the rate condemnations of cattle. Farm audits conducted by veterinary authorities or even retail auditors verify compliance with cattle health, welfare, food safety policy, and implementation of good agricultural practices, reducing the likelihood of total carcass condemnation at the slaughterhouse. 

The season (autumn and winter) has been described as a risk factor for cattle condemnation. Some authors [53] suggest that cattle grow more slowly due to certain health issues. Although the author does not specify the type of health conditions, there is an increased risk of bovine respiratory syndrome in cold season as referred elsewhere [67]. During the winter, the food is generally of lower quality (less forage availability and/or adequate volumes of pasture), mainly in extensive management, which could constitute a risk factor for the appearance of diseases. In addition, having cattle at barns during the winter can increase the risk of transmission of respiratory diseases and consequently the probability of condemnations by respiratory problems.

Regarding the price, it has been reported that in situations of greater demand for beef, where there is a higher price, farmers send more cattle to the slaughterhouses, including those animals of lower quality [53]. Regarding age, older cattle were more likely to be condemned for chronic diseases (i.e., chronic mastitis, reproductive problems, cachexia, etc.) [53,68]. After the bibliographic review, it has also been observed that cattle from farms with a higher mortality rate have a higher probability of cattle carcass condemnation in the slaughterhouse [68], probably related to aspects, such as poor welfare conditions, lack of disease control, and/or due to a lack of good agricultural practices, as described above.

The organ condemnation rate in cattle represents an important source of information on the infectious and parasitic diseases status of the herd. As previously indicated, the research available is scarce, out of date, and from local studies. 

Based to the published data (Table 3), the most condemned organ were the liver (35%, ranged from 3.6% to 73.75%) followed by lungs (7%, ranged from 5% to 55%), kidneys (5%, ranged from 0.16% to 8.74%), hearth (3.9%, ranged from 0.27% to 10.66%), intestines (4.75%, ranged from 2.18% to 7.31%), spleen (3.2%, ranged from 0.22% to 9.40%), and tongue (1.02%, ranged from 0.22% to 2.3%). However, lung condemnation was reported as the main cause in cattle [54]. 

The liver was mainly condemned due to the presence of parasitic lesions related to *Fasciola hepatica* and hydatidosis (*Echinococcus granulosus*). Condemnation for cirrhosis is expected since chronic trauma by parasitic diseases leads to the degeneration of liver parenchyma. These results highlight that prophylactic deworming programs are not a common practice in developing countries. 

**Table 3 vetsci-10-00167-t003:** Causes of condemnation of bovine organs.

Organ	Prev (%)	Cause	Country	Reference
Liver	17.58	Calcification (0.13%)	Ethiopia	[69]
		Cirrhosis (2.01%)		
		*Cysticercus bovis* (2.55%)		
		Hydatic cyst (3.62%)		
		Fasciolosis (9.26%)		
	31.2	Abcesses (23.2%)	Italy	[54]
		Distomatosis (23.7%)		
		Hidatidosis (6.80%)		
		Other causes (35.63%)		
		Perihepatitis (10.17%)		
	34.32	Abcess (6.20%)	Ethiopia	[70]
		Calcificartion (7.89%)		
		Cirrhosis (15.41%)		
		Faciolosis (48.5%)		
		Hepatitis (4.70%)		
		Hydatidosis 17.92%)		
	43.00	Calcified cyst (18.00%)		
		Cirrhosis (5.00%)		
		Fasciolosis (70.00%)	Tanzania	[57]
		Hydatidosis (3.00%)		
		Peritonitis (4.00%)		
Liver	61.64	Abscess (0.46%)	Ethiopia	[71]
		Calcification (4.16%)		
		Cirrhosis (10.4%)		
		Fasciolosis (7.4%)		
		Hepatitris 2.08%)		
		Hydatid cyst (17.1%)		
	7.20	Abcessess (2.08%)	Tanzania	[60]
		Calcified cyst (16.86%)		
		Fasciolosis (74.78%)		
		Hydatidosis (3.27%)		
		Peritonitis (2.98%)		
	73.75	Abcess (3.16%)	Ethiopia	[72]
		Calcification (20.78%)		
		*C. bovis* (3.16%)		
		Cirrhosis (1.82%)		
		Fasciola (40.70%)		
		Hematoma (2.19%)		
		Hepatization (2.67%)		
		Hydatic cyst (25.03%)		
	3.60	Abscess (7.40%)	Sudan	[73]
		Adhesion (0.20%)		
		Calcification (3.10%)		
		Cirrhosis (0.10%)		
		Congestion (0.30%)		
		*Cysticercus bovis* (13.50%)		
		Fasciola (51.60%)		
		Fatty change (2.40%)		
		Fibrosis (1.70%)		
		Hemorrhage (0.70%)		
		Hydatic cyst (0.30%)		
		Necrosis (18.60%)		
		Tuberculosis (0.10%)		
	18.63	Abscess (6.50%)	Tanzania	[58]
		Calcified cyst (10.63%)		
		Fasciolosis (48.81%)		
		Hydatidosis (18.26%)		
		Others ^1^ (15.78%)		
	25.7	Abcess (2.33%)	Ethiopia	[74]
		Adhesion (1.90%)		
		Calcification (27.54%)		
		Cirrhosis (5.61%)		
		Fasciolosis (37.50%)		
		Hydatidosis (25.10%)		
	18.75	Abcess (16.67%)	Ethiopia	[75]
		Calcification (16.67%)		
		Cirrhosis (20.83%)		
		Fasciolosis (33.33%)		
		Hydatic cyst (12.50%)		
	38.00	Abscess (4.38%)	Ethiopia	[76]
		Calcification (12.28%)		
		Cirrhosis (17.54%)		
		Fasciolosis (43.85%)		
		Hydatid cyst (21.92%)		
Liver	44.5	Abcess (3.50%)	Ethiopia	[77]
		Calcification (8.80%)		
		Cirrhosis (16.40%)		
		Fasciolosis (47.40%)		
		Hydatodosis (21.00%)		
		Necrosis (2.90%)		
	53.3	Abcesses (6.82%)	Ethiopia	[78]
		Calcification (19.02%)		
		Cirrhosis (5.83%)		
		Fasciolosis (42.92%)		
		Hydatodosis (22.43%)		
	34.25	Abscess	Ethiopia	[79]
		Cirrhosis		
		Fasciolosis		
		Fibrosis		
		Hydatidosis		
	50.3	Fasciolosis (51.01%)	Ethiopia	[59]
		Hepatitis (26.91%)		
		Echinoccocosis (7.37%)		
		*C. bovis* (6.83%)		
		Abcess (5.97%)		
		Tuberculosis (0,51%)		
		Calcification (0.33%)		
		Haemorraghe and hematomas (0.13%)		
		Tumour (0.11%)		
Lungs	64.86	Bronchopneumonia (14.62%)	Italy	[54]
		Pleurisy (51.20%)		
		Pneumonia (19.74%)		
	8.19	Abcessess (0.40%)	Ethiopia	[69]
		Emphysema (1.61%)		
		Hydatic cyst (5.1%)		
		Pneumonia (1.07%)		
	19.68	Abcess (19.34%)	Ethiopia	[70]
		Hydatic cyst (68.20%)		
		Pneumonia (12.45%)		
	19.00	*Cryptosporidium bovis* (2.00%)	Ethiopia	[57]
		Emphysema (22.00%)		
		Hydatodosis (6.00%)		
		Pleurisy (14.00%)		
		Pneumonia (52.00%)		
		Tuberculosis (4.00%)		
	26.29%	Abscess (3.00%)	Ethiopia	[71]
		Emphysema (10.40%)		
		Fibrosis (7.50%)		
		Hepatisation (16.40%)		
		Hydatic cist (53.70%)		
		Pneumonia (9.00%)		
Lungs	10.52	Abcessess (0.15%)	Tanzania	[60]
		Calcified cyst (2.75%)		
		Congestion (0.01%)		
		Emphysema (32.78%)		
		Haemorrhages (21.86%)		
		Hydatidosis (27.27%)		
		Pleurisy (6.60%)		
		Pneumonia (8.51%)		
		Tuberculosis (0.03%)		
	14.14	Abcess (6.25%)	Ethiopia	[72]
		Emphysema (5.63%)		
		Haemorrhage and hematoma (58.75%)		
		Hydatid cyst 12.50%)		
		Pneumonia (16.88%)		
	14.14	Abcess (6.25%)	Ethiopia	[72]
		Emphysema (5.63%)		
		Haemorrhage and hematoma (58.75%)		
		Hydatid cyst 12.50%)		
		Pneumonia (16.88%)		
	13.24	Abcessess (8.19%)		[58]
		Antracosis (7.29%)		
		Calcified cyst (9.03%)		
		Emphysema (13.07%)		
		Hydatodisis (22.19%)		
		Melanosis (2.89%)		
		Pleurisy (6.44%)		
		Pneumonia (30.13%)		
		Tuberculosis (0.71%)		
	24.8	Abccess (0.76%)	Ethiopia	[74]
		Calcification (1.97%)		
		Cysticercus bovis (0.32%)		
		Distomatosis (1.42%)		
		Emphysema (0.87%)		
		Hydatodosis (86.60%)		
		Pneumonia (7.78%)		
Lungs	6.51	Abscess (36.00%)	Ethiopia	[75]
		Emphysema (24.00%)		
		Hydatic cyst (16.00%)		
		Pneumonia (24.00%)		
	32.7	Abscess (4.08%)	Ethiopia	[76]
		Calcification (7.65%)		
		Congestions (9.18%)		
		Emphysema (11.22%)		
		Granulomatous lesion (1.53%)		
		Hydatic cyst (62.24%)		
		Pneumonia (4.08%)		
Lungs	35.7	Abscess (4.40%)	Ethiopia	[77]
		Emphysema (28.50%)		
		Hepatisation (2.90%)		
		Hydatidosis (35.70%)		
		Pneumonia (28.50%)		
	55.5	Abscess (3.25%)	Ethiopia	[78]
		Calcification (15.96%)		
		Congestion (3.72%)		
		Emphysema (7.98%)		
		Haemmorrage (2.79%)		
		Hydatid cyst (64.31%)		
		Pneumonia (1.86%)		
	16.46	Abscessation (5.68%)	Ethiopia	[79]
		Emphysema (7.95%)		
		Fibrosis (3.40%)		
		Hydatidosis (73.86%)		
	35.6	Abscess (2.60%)	Ethiopia	[59]
		Echinoccocosis (26.97%)		
		Emphysema (26.00%)		
		Pneumonia (43.00%)		
		Tuberculosis (3.97%)		
		Tumor (0.06%)		
Kidneys	1.21	Hydatic cyst (0.27%)	Ethiopia	[69]
		Hydronephosis (0.94%)		
	1.1	Hydatidosis (29.40%)	Ethiopia	[70]
		Hydronephitis (17.65%)		
		Infarcts (23.53%)		
		Pyonephitis (29.40%)		
	25.00	Congestion (3.00%)	Tanzania	[57]
		Cyst (7.00%)		
		Hydronephosis (17.00%)		
		Infarcts (58.00%)		
		Nephitis (15.00%)		
	8.74	Hydatic cyst (40.00%)	Ethiopia	[71]
		Hydronephrosis (60.00%)		
Kidneys	6.27	Cysts (23.11%)	Tanzania	[58]
		Fatty change (11.55%)		
		Hydronephosis (29.60%)		
		Infarct (12.78%)		
		Melanosis (5.73%)		
		Nephritis (17.20%)		
	0.16	*Cysticercus bovis* (60.00%)	Ethiopia	[74]
		Hydatidosis (40.00%)		
	3.9	Congenital cysts (52.99%)	Tanzania	[60]
		Hydronephrosis (34.02%)		
		Infarcts (5.62%)		
		Renal calculi (7.31%)		
	1.25	Abscess (42.86%)	Ethiopia	[72]
		Atrophy (28.57%)		
		Nephritis (28.47%)		
Kidneys	6.5	Haemorrahage (24.00%)	Ethiopia	[77]
		Infarcts (12.00%)		
		Nephritis (64.00%)		
	0.56	Fibrosis (100%)	Ethiopia	[79]
	4.39	-	Italy	[54]
Heart	0.27	*Cysticercus bovis* (100%)	Ethiopia	[69]
	2.06	Abscess (12.50%)	Ethiopia	[70]
		Hydatidosis (9.40%)		
		Pericarditis (78.10%)		
	3.09	Hydatic cyst (42.90%)	Ethiopia	[71]
		Pericarditis (57.10%)		
	0.73	Calcifued cyst (56.82%)	Tanzania	[60]
		Pericarditis (38.73%)		
		Cysticercosis (4.48%)		
	10.66	Abscess (4.20%)	Ethiopia	[72]
		*C. bovis* (8.40%)		
		Hemorrhage and hematoma (22.69%)		
		Hydatic cyst (22.69%)		
		Pericarditis (4.20%)		
	7.00	Calcified cyst (33.00%)	Tanzania	[57]
		*Cristosporidium bovis* (17.00%)		
		Degenration (8.00%)		
		Emaciation (3.00%)		
		Flabby hearts (2.00%)		
		Pericarditis (37.00%)		
Heart	2.98	Calcified cysts (29.44%)	Tanzania	[58]
		*Cysticercus bovis* (4.35%)		
		Haemorrhages (22.94%)		
		Hydatidosis (5.04%)		
		Melanosis (3.13%)		
		Pericarditis (35.07%)		
	2.7	*Cysticercus bovis* (33.66%)	Ethiopia	[74]
		Hidatidosis (8.92%)		
		Pericarditis (57.42%)		
	4.43	Abscess (29.41%)	Etiopía	[75]
		Edema (35.29%)		
		Hydatic cyst (35.29%)		
	4.7	*Cysticercus bovis* (28.57%)		[76]
		Hydatid cyst (50.00%)		
		Pericarditis (21.42%)		
	6.8	Edema (42.3%)	Ethiopia	[77]
		Pericarditis (42.3%)		
		Petechial haemorrhage (15.4%)		
	1.67	*Cysticercus bovis* (11,1%)	Ethiopia	[79]
		Hydatidosis (88.2%)		
	3.70	-	Italy	[54]
Tongue	0.22	*Cysticercus bovis* (100)	Ethiopia	[71]
	2.3	Abscess (56.6%)	Ethiopia	[76]
		*C. bovis* (0.56%)		
		*Cysticercus bovis* (44.4%)		
	0.56		Ethiopia	[79]
	0.16	-	Italy	[54]
Intestines	7.31	Abcessess (0.17%)	Tanzania	[60]
		Enteritis (20.63%)		
		Peritonitis (1.14%)		
		Pimply gut (78.04%)		
	2.18	Enteritis (49.09%)	Tanzania	[58]
		Pimply guts (50.91%)		
Stomachs	11.63	Adherences (41.69%)	Italy	[54]
		Foreign body lesion (41.17)		
		Other causes (17.14%)		
Spleen	0.22	Abcesses (4.03%)	Tanzania	[60]
		Peritonitis (4.02%)		
		Splenomegaly (91.95%)		
	2.05	Abcess 30.46	Tanzania	[58]
		Haematoma 14.85		
		Hydatidosis 31.81		
		Splenomegaly 22.86		
	1.8	Hydatic cyst (72.70%)		[76]
		Splenomegaly (27.27%)		
	0.01	Splenitis (69.40%)	Italy	[54]
		Splenomegaly (30.60%)		

^1^ Others include hepatitis, fatty degeneration, melanosis and cirrhosis.

Regarding the lungs, inflammatory diseases of the pleura and parenchyma, as well hydatid disease, represent the main causes of condemnation. These lesions are expected according to the management of cattle described in the literature that it is based on extensive management. Although bovine tuberculosis has been referred to as one of the most important cause of carcass condemnation, lung condemnation due to the presence of tuberculous lesions has not been described. The thoracic cavity is known to be the main location of bovine tuberculosis lesions [80]. These results may be biased due to a lack of cattle characterization and sampling methods (i.e., no data regarding age are available). 

Hydronephrosis and cysticercosis have been reported to be the main causes of kidney condemnation in cattle. Hydronephrosis is defined as a distension of the renal pelvis and calyx leading to loss of renal function [81]. The main cause is related to obstruction of the urinary tract due to urolithiases, tumors, or eversion of the urinary bladder [82]. Cysticercosis was reported as one the main causes, suggesting that obstruction of urinary tract could be associated with the presence of cyst. However, it appears that the kidneys are not main site of infection for *C. bovis* [83]. 

The main causes of heart condemnation are related to pericarditis and *C. bovis*. Pericarditis is usually secondary to respiratory problems [84]. Since the data presented in Table 3 are related to beef cattle rear in extensive conditions, this may explain the high rate of condemnation described in the literature. Although the presence of hydatic cyst at heart has been reported [85], other research only describes *Theileira* spp. and *Neospora caninum* [86,87] as causes of heart disease of parasitic origin in cattle. These differences are probably related to the country of origin in which the studies were conducted. Likewise, the presence of hydatid cysts in the heart has been described as atypical location [88]. Although other causes of hearth condemnation, such as lymphoma or traumatic reticulopericarditis, have been described [84], the presence of *C. bovis* cyst in hearth was not reported as a major cause.

### 2.2. Surveillance of Small Ruminant Carcass and Organs Condemnation

Reports of condemnations of small ruminant carcasses and organ are scarce as described above for cattle. They are usually from local studies, not updated and with variable results among them due to sampling and data analysis. The condemnation rate is much more variable than cattle, ranging from 0.01% to 7.2% (average 3.3%) (Table 4). In addition, the main causes of small ruminant condemnation seem to vary between studies and geographical location. Cachexia/wasting is mentioned as the main cause of carcass condemnation in developing countries [54], while *Sarcocystic* spp. cyst were referred to in developed countries [89]. Others reports [90] indicate a carcass condemnation rate of approximately 5% in sheep with the presence of multiple abscesses related to maedi-visna, paratuberculosis, and caseous lymphadenitis. Other emerging diseases, such as *Anaplasma ovis*, displayed a condemnation rate of approximately 35% due to jaundice [91].

However, these results must be carefully considered since there is not any international recognized classification of small ruminant lesions available. 

Research on the total and partial small ruminant condemnation and their causes worldwide is presented in Table 5. The main causes of liver and spleen condemnation are related to parasitism, such as *F. hepatica*, *C. tenuicolis*, and *Echinococcus* spp. (hydatic cyst), while pneumonia and emphysema are related as the main causes of pulmonary condemnation. These causes of organ condemnation are expected since most available studies come from countries in which extensive management predominates. The few prophylactic deworming programs may explain the high rate of liver and spleen condemnation by parasitism. However, other lesions, such as cirrhosis or infectious necrotic hepatitis derived from chronic parasitism, do not seem to be relevant. In addition, parasitic problems associated with the extensive management (probably including periods of lack of pasture/feed in some seasons) makes small ruminants more susceptible to infectious diseases, which may explain the high rate of lung condemnation by respiratory lesions. These results are in accordance with those reported elsewhere [92] indicating respiratory problems as the main cause of lamb death. 

**Table 4 vetsci-10-00167-t004:** Small ruminant causes of carcass condemnation.

Species	TC (%)	Cause of Condemnation (%)		Country	Reference
Sheep	-	Abscess	0.02	Tanzania	[55]
		Arthritis	0.03		
		Bruising	0.07		
		Emaciation	0.93		
		Pneumonia	0.07		
		Pyaemia	0.13		
		Septicaemia	0.13		
		Tumours	0.01		
S&G	0.009	Jaundice	35.56	Italy	[54]
		Other causes	20.00		
		Peritonitis	44.44		
S&G	0.02	-	-	Iran	[56]
Sheep	6.35	-	-	Ethiopia	[93]
Sheep	0.063	Abscesses	71.8	Tanzania	[58]
		Emaciation	17.9		
		Jaundice	10.3		
Goats	0.106	Abcesses	43.6		
		Emaciation	34.7		
		Jaundice	21.7		
Sheep	6.42	Abcess/pyemia	7.8	United States	[94]
		Arthritis	3.3		
		Carcinoma	0.3		
		Caseous lymphadenitis	18.8		
		Coccidiodal granuloma	0.0		
		Contamination	0.5		
		Cysticercosis	11.6		
		Emaciation	6.8		
		Eosinophilic myosistis	3.0		
		Gen. miscellaneous	0.9		
		Icterus	9.9		
		Injuries	0.5		
		Malignant lymphoma	0.3		
		Mastitis	0.0		
		Metritis	0.1		
		Misc. infectious dis.	0.1		
		Misc. inflame. Dis.	0.3		
Sheep	6.42	Misc. Neoplasm	0.4	United States	[95]
		Misc. parasitism	2.6		
		Nephritis/pyelitis	1.4		
		Non ambulatory	0.1		
		Other reportable dis.	2.5		
		Pericarditis	0.5		
		Peritonitis	1.7		
		Pigmentary condition	0.1		
		Pneumonia	9.2		
		Residue	0.2		
		Sarcoma	0.3		
		septicemia	5.2		
		Toxaemia	6.7		
		Uremia	3.2		
Sheep	6.7	-		Ethiopia	[96]
Goat	7.2				
Sheep	0.10	-		Palestine	[52]

Misc: miscelaneous, Dis. TC.: total condemnation; S&G: sheep and goats.

**Table 5 vetsci-10-00167-t005:** Cause of organs condemnation in small ruminants.

Organ	Species (% ^a^)	Cause of Condemnation (% ^b^)	Country	Reference
Liver	Sheep (18.7)	Abcessess (5.7)	Tanzania	[58]
		Abcessess (8.58)		
		Calcified cyst (8.19)		
		Calcified cysts (6.3%)		
		*Cisticercus tenuicolis* (1.4)		
		*Cysticercus tenuicolis* (2.20)		
		Faciolosis (18.2)		
		Fasciolosis (17.09		
		Hydaidosis (19.18)		
		Hydatidosis (20.4)		
		Other (0.4) *		
		Stilesiosis (44.72%)		
		Stilesiosis (47.6)		
	Sheep (77.15)	-	Italy	[54]
	Sheep (61.23)	*C. teniculosis* (18.77)	Ethiopia	[96]
		Calcification (49.78)		
		Cirrosis (7.65)		
		Fasciolosis (9.36)		
		Hepatitis (3.82)		
		Hidatic cyst (2.97)		
		*S. hepatica* (7.65)		
	Goat (42.19)	*C. teniculosis* (30.27)	Ethiopia	[96]
		Calcification (45.67)		
		Cirrosis (5.55)		
		Fasciolosis (2.46)		
		Hepatitis (7.40)		
		Hydatic cyst (0.60)		
		*S. hepatica* (8.05)		
	Goats (17.91)	Abcessess (8.6)	Tanzania	[58]
		Calcified cyst (10.3)		
		*Cysticercus tenuicolis* (1.5)		
		Fasciolosis (17.2)		
		Hydatidosis (21.3)		
		Stilesiosis (41.1)		
	Sheep (58.5)	Abscess (3.85)	Ethiopia	[95]
		*C. tenuicolis* (9.05)		
		Calcifications (8.90)		
		Cirrhosis (5.34)		
		Fasciolosis (11.86)		
		Hepatitis (30.11)		
		Hydatid cyst (1.48)		
		Mechanical damage (10.58)		
		Other causes (2.81)		
		*Stelesia hepatica* (16.02)		
Liver	Goat (43.8)	Abscess (5.64)	Ethiopia	[95]
		*C. tenuicolis* (18.87)		
		Calcifications (9.55)		
		Cirrhosis (5.20)		
		Fasciolosis (8.17)		
		Hepatitis (8.91)		
		Hydatid cyst (4.01)		
		Mechanical damage (9.80)		
		Other causes (2.22)		
		*Stelesia hepatica* (27.63)		
Lungs	Sheep (7.85)	Abscesses (14.3)	Tanzania	[58]
		Calcified cyst (17.2)		
		Emphysema (17.9)		
		Hydatodosis (19.2)		
		Pneumonia (31.4)		
	Goats (8.43)	Abcesses (16.1)	Tanzania	[58]
		Calcified cyst (15.3)		
		Emphysema (17.8)		
		Hydatodosis (17.2)		
		Pneumonia (33.6)		
	Sheep (3.80)	-	Italy	[56]
	Sheep (44.5)	Abscess (5.06)	Ethiopia	[95]
		Calcification (6.04)		
		Emphysema (15.39)		
		Hydatid cyst (7.44)		
		Others (3.11)		
		Pneumonia (62.96)		
	Goat (41.7)	Abscess (5.00)	Ethiopia	[95]
		Calcification (5.63)		
		Emphysema (16.56)		
		Hydatid cyst (6.40)		
		Others (2.97)		
		Pneumonia (63,44)		
	Sheep (77.86)	Abscess (6.02)	Ethiopia	[96]
		Calcification (1.00)		
		Emphysema (24.41)		
		Hydatid cyst (4.34)		
		Marbling (2.00)		
		Pneumonia (62.20)		
	Goats (78.39)	Abscess (1.66)	Ethiopia	[96]
		Calcification (1.66)		
		Emphysema (18.27)		
		Hydatid cyst (0.66)		
		Marbling (5.98)		
		Pneumonia (71.76)		
Spleen	Sheep (0.33)	Abcesses (47,5)	Tanzania	[58]
		Hydatidosis (52.5)		
	Goat (0.68)	Hydatidosis (70.3)		
		Abcesses (29.7)		
Heart	Sheep (8.6)	Abscess (3.03)	Ethiopia	[95]
		*C. ovis* (5.05)		
		Calcification (11.11)		
		Other (17.18)		
		Pericarditis (63.63)		
	Sheep (0.05)	-	Italy	[54]
	Goat (7.5)	Abscess (5.24)	Ethiopia	[95]
		*C. ovis* (6.08)		
		Calcification (13.91)		
		Other (11.30)		
		Pericarditis (63.47)		
	Sheep (9.92)	Calcification (28.94)	Ethiopia	[96]
		Hydropericardium (7.89)		
		Pericarditis (63.15)		
	Goat (8.59)	Calcification (42.42)	Ethiopia	[96]
		Pericarditis (48.48)		
		Hydropericardium (9.09)		
Kidney	Sheep (14.32)	Abscess (16.36)	Ethiopia	[96]
		Nephrosis (14.54)		
		Nephritis (69.10)		
	Goat (14.32)	Abscess (5.34)	Ethiopia	[96]
		Nephrosis (41.33)		
		Nephritis (53.33)		
	Sheep (0.05)	-	Italy	[54]

* Other causes include telangiectasis, hepatitis, fatty degeneration, melanosis and liver cirrhosis. ^a^ Represents the total % of organ condemnations. ^b^ Represent the % of cause of condemnation in each organ.

### 2.3. Surveillance of Swine Carcass and Organs Condemnation

There is more information on the condemnation of pig carcasses than cattle and small ruminant probably associated with the need of knowledge of pathological processes for risk-based inspection [97]. The rate of pig carcass condemnation is about 0.37% (ranged from 0.1% to 0.57%) although other works reported condemnation values higher than 8.5% [98] and up to 10% [99].

Variations in the causes and prevalence of carcass condemnations (Table 6) reported in different studies may be associated with the geographical area, climatic conditions, farm management, and herd health status [98,100]. Furthermore, differences in the terminology of swine inspection results may also influence the condemnation rate [101].

Osteomyelitis [23], abscesses [98], erysipelas, generalized jaundice [39], arthritis [102], contamination by eviscerating leaking [99], or sensory changes in meat [103] have been reported as main causes of carcass condemnation. Other reports [104] indicated mange as the main cause of carcass condemnation while abscesses and peritonitis accounted up to 50% of them. 

**Table 6 vetsci-10-00167-t006:** Causes of pig carcass condemnation.

CR (%)	Cause of Condemnation (%)	Country	Reference
0.24	Abscesses (8.42)	Portugal	[23]
	Bloody meat (0.26)		
	Caquexia (1.79)		
	Erysipela (0.77)		
	Febrile meat (0.26)		
	Generalised melanosis (1.79)		
	Granulomatous lymphadenitis (22.70)		
	Jaundice (0.26)		
	Muscular necrosis (0.26)		
	Osteomyelitis (38.52)		
	Pale soft and exudative (PSE meat) (0.51)		
	Peritonitis (2.55)		
	Pleurisy/pneumonia (21.17)		
	Purulent Metritis (0.51)		
	Purulent nephritis (0.26)		
8.5	Abcesses (55.80)	Spain	[98]
	Arthritis (7.40)		
	Cachexia (28.90)		
	Catarrhal bronchopneumonia (16.20)		
	Erysipelas (1.20)		
	Fibrous peritonitis (6.40)		
	Fibrous pleuritis (6.40)		
	Jaundice (3.50)		
	Pleuropneumonia (5.50)		
	Putrid meat (1.00)		
	Tail lesions (2.90)		
	Vertebral osteomyelitis (9.60)		
0.10	Anaemia (10.66)	Spain	[97]
	Arthritis, osteomielitis (4.20)		
	Ascaridiasis (0.15)		
	Contamination (1.05)		
	Cryptosporidiosis (0.15)		
	Emaciation (16.82)		
	Erisipelas (7.36)		
0.10	Generalized (pyemias) (34.08)	Spain	[97]
	Haemorrhages, edemas (0.60)		
	Insufficient bleeding (4.35)		
	Jaundice (3.45)		
	Melanomas (0.15)		
	Metritis (0.15)		
	Odour (0.15)		
	PDNS (4.20)		
	Pericarditis (0.60)		
	Peritonitis (3.45)		
	Pleuropneumonia (5.86)		
	Ptyriasis rosea (0.15)		
	Pyelonephritis (0.90)		
	Sarcosporidiosis (0.30)		
	Tuberculosis (1.20)		
0.03	Abscesses (7.53)	Italy	[39]
	Cachexia (2.69)		
	Disseminated hemorrhagic síndrome (0.54)		
	Enteritis (5.38)		
	Errors in the slaughtering process (1.34)		
	Erysipelas (37.36)		
	Generalized jaundice (26.07)		
	Lipomatous pseudohypertrophy (9.95)		
	Neoplasia (0.54)		
	Perihepatitis (1.07)		
	Peritonitis (3.70)		
	Pleuritis (2.15)		
	Polyserositis (0.54)		
	PSE (0.27)		
	Septicemia (0.27)		
	Traumatic lesions (0.54)		
0.37	Arthritis (8.31)	Canada	[102]
	Enteritis (2.86)		
	Nephitis (6.94)		
	Other (76.42)		
	Pneumonia (5.46)		
1.40	-	Brazil	[103]
0.17	-	Italy	[54]
0.57	Added deleterious substances (8.31)	Czech Republic	[105]
	Boar taint (4.33)		
	Digestive infections (0.31)		
	Miscellaneous infections (2.02)		
	Non-infectious diseases (20.57)		
	Parasitic diseases (0.05)		
	Respiratory infections (6.03)		
	Salmonella infection (0.03)		
	Sensorial changes in meat (58.23)		
	Tuberculosis infection (0.08)		
0.11	-	Portugal	[106]
10.20	Abscess (0.580)	Brazil	[99]
	Adherences (3.72)		
	Contamination by eviscerating leaking (1.79)		
	Cryptorchidism (0.149)		
	Excessive scalding (0.12)		
	Lymphadenitis (0.29)		
	Peritonitis (0.10)		
	Pleurisy (0.85)		
	Pneumonia (0.20)		
	Scabies (0.14)		
	Suppurated wounds (0.13)		
	Traumatic lesion (1.57)		
0.3	-	Finland	[107]

CR: condemnation rate.

Vertebral osteomyelitis has been defined as an inflammation of the vertebrae with involvement of the medullar cavity, generally secondary to bacterial infections or skin trauma [108]. The total condemnation is based on septicemia caused by the dissemination of pyogenic bacteria. Vertebral osteomyelitis is the main condemnation of pig carcass in Portugal [23,105,109]. Tail biting has been described as a predisposing factor for osteomyelitis in pigs [110,111]. This behavior has been widely studied as an indicator of reduced animal welfare, although it may be influenced by various external and internal factors such as the environment, feeding, housing, male-to-female ratio, genetics, sex, or age, among others [112,113]. Mycobacterial infection in swine caused by *Mycobacterium avium* subsp *hominisuis*, *M. avium* subsp. *avium*, *M. intracellulare*, all belong to the *Mycobacterium avium* complex (MAC), which represents the main cause of granulomatous lymphadenitis in slaughtered pig at post-mortem examination [114,115,116]. However, other nontuberculous microorganisms, such as *Trueperella pyogenes*, *Rhodococcus equi*, and *Streptococcus* spp., may contribute to granulomatous lesions [117]. Swine infections caused by MAC result in severe economic losses for producers due to carcass condemnation.

Although MAC disease does not usually present clinical signs, some disorders have been associated [118]. The lesions, generally developed in the lymph nodes of the head and/or the mesentery [119], and the isolation of an etiological agent have been described by several authors [115,120].

Control of granulomatous lymphadenitis in the slaughterhouse is of major public health importance as *Mycobacterium* spp. has the ability to develop lung disease, lymphadenitis in children, or septicemia in immunocompromised patients [121]. Given that MAC is distributed worldwide along with the little research on its prevalence in pig farms [122,123], it suggests the need for a careful risk assessment regarding the implementation of visual-only meat inspection [124]. Jaundice, defined as the accumulation of bilirubin, bile pigments, or hemoglobin in the blood, presents as an intense yellow hue in carcass and organs. 

The condemnation of carcasses due to generalized jaundice is associated with a systemic infection (toxic or bacteriological), such as leptospirosis, *Mycoplasma suis*, or *Salmonella cholerasuis*, among others. 

Erysipelas has also been reported as an important cause of pig carcass condemnation [39]. Erysipelas is an infectious disease caused by the bacterium *Erysipelothrix rhusiopathiae* that caused characteristic diamond-shaped erythematous skin lesions. *E. rhusiopathiae* is a bacterium with a very wide host range (also zoonotic) and common in pigs, especially between 2 and 12 months of age. Its condemnation is based on its septicemic condition.

Regarding pulmonary problems, a lower condemnation rate is observed in pigs than cattle and small ruminants. Although respiratory problems are one of the main health problems in livestock intended for meat production, the lower condemnation rate could be explained by the fact that the pigs are raised in farms with controlled environmental conditions, such as temperature and humidity [125].

The factors related to condemnation of carcasses are not yet clear. Reports indicate higher condemnation rate in winter than summer or autumn [126], probably explained because pigs born during spring and early summer are subject to less environmental stress. However, other authors [99] concluded that month and season were not significantly associated with carcass condemnation. Furthermore, it has been reported [126] that the density of swine farms in a geographical area influences the rate of carcass condemnation associated with local disease outbreaks. 

Carcass condemnation due to generalized melanosis, osteomyelitis, and granulomatous lymphadenitis appear to be associated to the season [23]. Regarding osteomyelitis, the odds of condemnation were lower after the compulsory fulfilment of the food chain information form. However, its compliance does not influence other causes of carcass condemnation [23].

The improvement in animal welfare on farms in recent years may explain the decrease in condemnations due to osteomyelitis. Since infection of pigs by MAC is probably environmental, the influence of year may be related to regional characteristics, such as weather conditions and/or presence of hosts [127].

Summer and autumn seem to have an influence on the condemnation of the carcass due to anemia [97] probably associated with the seasonality of the digestive process. Moreover, condemnations due to insufficient bleeding seem to be higher in summer, probably related to an increase in pigs fatigued due to higher temperatures. 

Light carcass weight represents a risk factor for carcass condemnation of a sow at the end of the productive cycle, probably associated with lower weight gain due to lower intake rates [100]. 

Regarding partial condemnation (Table 7), scarce data are available. Liver, kidney, and heart are the most condemned organs, mainly by parasitic lesions [128]. However, abscesses still represent an important rate of partial condemnation and an economic issue [129]. Moreover, the presence of specific post-mortem lesions, such as arthritis, shoulder ulcers, or pneumonia, was associated with sow mortality, highlighting the importance of data surveillance at the slaughterhouse to monitor herd health [106].

## 3. Surveillance of Animal Welfare in the Slaughterhouse

Animal welfare is a growing concern for consumers who demand not only healthier and safer food, but also food obtained through practices that ensure adequate animal protection [130]. In addition, animal welfare is part of the European Union farm-to-fork strategy, which aims to make Europe’s agricultural practices more sustainable through an integrated food policy that encompasses the entirety of the European Union supply chain [131]. As part of this strategy, the EU is carrying out a comprehensive evaluation of its animal welfare legislation, for which the European Food Safety Agency (EFSA) has been asked to provide new opinions reflecting the most up-to-date research and scientific data to address animal welfare, on the farm, during transport, and in the slaughterhouse [132,133,134]. The evaluation of the animal welfare through specific parameters in the slaughterhouse is a difficult task since there is not a clear consensus on which parameters must be evaluated.

In addition, when the animals are destined to the slaughterhouse, they are subjected to new stimuli that can modify their well-being. Therefore, the Farm Animal Welfare Council (FAWC) has proposed the use of some animal-based indicators to assess the general welfare of animals [101,135], to assess the existence of welfare problems both on the farm and in the stages prior to slaughter. Slaughterhouses are ideal places to assess animal welfare since we can observe various indicators of various animal species, from large geographical areas, and we can observe the variations of these indicators over time [136]. Some welfare indicators have been reported in the literature, such as endocrine measures (e. g. plasma cortisol concentration), observation of specific behaviors (e.g., ruminating) or pain responses (e. g. nostril stimulation). However, their measurement in the slaughterhouse is not even possible due to high economical cost, impossibility to test all animals, and interference with the speed of the slaughter line. Instead, it has been suggested that indicators related to physiological aspects, morphometric characteristics, animal behavior, and quality, are welfare indicators that can be easily in the slaughterhouse with minimal impact on the speed of slaughter line [137]. It is important to refer that the surveillance of welfare indicators described below is not currently mandatory in the EU. In addition, this surveillance implies the need for (at least) one more OVI dedicated exclusively to welfare evaluation. 

### 3.1. Surveillance of Cattle Welfare at Slaughterhouse

Bruises are defined as lesions where tissues are crushed with a rupture of the vascular supply and an accumulation of blood and serum without discontinuity of the skin. Bruises have been used as an indicator of welfare because they provide information on pre-slaughter stages and handling of cattle. The prevalence of carcass bruising is variable among studies and influenced by both handling (loading and unloading operations, type of transport, road type, etc.) and animal factors (sex, age, breed, weight, horns, etc.) [63,65,138,139,140,141,142]. Other authors [63] showed an occurrence of bruises greater than 50%. Most bruises are characterized as small, circular or irregular in shape, and hemorrhagic. Since most of the bruises were considered recent events, it suggests that they are related to handling at the slaughterhouse. However, the presence of multiple carcass bruises associated with poor body condition score or poor cleaning of live cattle may suggest deficiencies in the quality of welfare on the farm. 

Other studies [65] reported a prevalence of bruises of 60%, indicating that most bruises, depending on their location, are related to handling and transport to the slaughterhouse. Similar results were reported elsewhere [143] in which bruises were influenced by the animal load at transport and the number of loading and unloading operations. It has also been reported [144] that slaughtered cattle from livestock markets had 1.5 times more bruising that cattle slaughtered directly from farm. The presence of linear bruises in cattle from markets indicates that cattle have been beaten more frequently. Since cattle size is bigger than small ruminants or pigs, movement of cows trough tout corridors, pens, and/or in loading and unloading operations is facilitated by farmers with the help of a stick [136]. The high prevalence of bruising on carcasses highlights the importance of welfare verification both on the farm and in the pre-slaughter phases.

Since most of the studies indicate that carcass bruises are recent events, they point to the need for proper equipment (i.e., trailers, restraining cages, etc.) and training for both farmers and slaughterhouse personnel (i.e., driving, loading, and unloading). 

The main problem lies in detecting and quantifying (older) lesions in the slaughterhouse that are indicative of poor welfare during the growth of fattening animals on the farm [145]. The use of infrared thermography (IT) has been suggested for the detection of ante-mortem bruising [146]. The main Di vantage of this technique is associated to the scarce detection of recent bruises (e.g., low sensitivity), such as those produced in transport, since the inflammatory response of the tissues only appears after at least 24 h [146].

A previous study that evaluated the welfare of cattle through individual indicators (including carcass bruises, hoof injuries and organ condemnations, among others) indicates the presence of severe bruising, hoof problems, and high liver condemnation, highlighting the existence of welfare and health problem likely due to a lack of prophylactic deworming, poor bedding, and/or poor hoof trimming [147]. 

The evaluation of the well-being of cattle using the health of hoof in the slaughterhouse [148] showed that most of the cattle showed hoof disorders (abnormal claw shape, 55%; fissures on the claw wall, 25%; skin wounds, 16%; and sole disorders, 15%). Since these lesions cause pain, discomfort, and affect the farm performance, these authors suggest that the evaluation of hoof disorders could be included as part of protocol for evaluating the welfare of cattle at the slaughterhouse.

The association between cattle welfare and microbiological quality of beef carcasses has also been indicated [149]. Although a non-significant association was found between cattle welfare indicators and microbiological quality, they concluded that *Salmonella* spp. and total bacterial counts were related with pH (*p* < 0.05) and the number of bruises in the carcasses (*p* < 0.05).

Other work [150] studied the influence of some welfare indicators in live cattle (including lameness, cleanliness, bruises, hair loss and body condition score) and its relationship with the hot carcass weight. The results showed that cattle with a higher number of movements in life, presence of bruises, and poor body condition had lower carcass weight. 

### 3.2. Surveillance of Swine Welfare at Slaughterhouse

The role of farm welfare and the presence of post-mortem lesions at the slaughterhouse in pigs were assessed [151]. Therefore, pigs reared in intensive farms displayed more respiratory lesions resulting in a high rate of lung condemnation. This fact could be associated to high concentrations of dust in the environment from the feed and to other environmental conditions on farms, such as relative humidity, ammonia concentrations, or variations in temperature that also influence the appearance of respiratory problems. According to the same author, the fact that the liver and kidneys are one of the most condemned organs (after the lungs) may be associated with the intensive rearing regimen with concentrated feeding and intensive metabolism.

The assessment of lameness, fear (reluctance to move), slips, and falls to monitor welfare during arrival at the slaughterhouse has been studied by interobserver reliability [152]. Although this work studied the best conditions to evaluate the welfare of pigs in the slaughterhouse, the results indicate that prevalence of lameness (3.22%) and fear (4.37%) are low, suggesting proper handling of the pig on the farm. Furthermore, lameness has been suggested as an indicator of welfare in pigs reared on unsuitable soils related to extensive management [153].

Another report [154] also evaluated some indicators of welfare in the slaughterhouse and, when possible, compare them with those collected at the farm. The authors found that dermatitis, white spot, wounds on the body (ear lesions), manure on the body and bursitis had a similar occurrence on both the farm and slaughterhouse. Therefore, the presence of manure on the body was related to the absence of clean and dry areas on the farm while ear injuries may be related to poor housing conditions. The study of several welfare indicators on both the farm and in the slaughterhouse [154] showed that the prevalence of bursitis was similar in both locations, whereas ear injuries and tail biting were more frequent on the farm and in the slaughterhouse, respectively, unlike previously reported [155].

Likewise, skin lesions, hernias or rectal prolapse presented similar values in both locations. However, the high variability in the prevalence of lesions suggests the existence of risk factors on the farm. Thus, it was observed that the lesions on the ears and tails were greater in sows. In addition, these authors indicate that the evaluation of welfare indicators is better evaluated at the slaughterhouse than in the farm, probably related to the greater number of pigs that were inspected at slaughter. However, aspects, such as large groups of animals lying down next to each other or the speed of the slaughter line, can make it difficult to properly assess the welfare of pigs both on the farm and in the slaughterhouse, respectively [153]. Currently, precision livestock farming technologies are used to integrate welfare information at farms. This information, together with data collected at slaughterhouse, can be used for farmers to improve welfare on farms. Additionally, this welfare information can be share with consumers to improve confidence about food of animal origin [156]. 

### 3.3. Surveillance of the Welfare of Small Ruminant in the Slaughterhouse

The evaluation of the welfare of small ruminants on farms has been carried out through a systematic literature review [137,154] based on the classification of each indicator according to the five freedoms [157]. As discussed above, the difficulty in assessing welfare at the slaughterhouse is determining which indicators should be considered [158]. It has been suggested that body cleanliness, carcass bruising, diarrhea, skin lesions, skin irritation, castration, notching the ears, tail docking, and the presence of small ruminant with an evident “sick condition” can be measured in the slaughterhouse [136]. However, the problem, also discussed above, is to establish a homogeneous benchmark for each indicator. 

As seen in cattle, carcass bruising is an indicator of animal welfare prior to slaughter [159]. The prevalence of bruises in sheep is low since the wool acts as a protector [160], although other authors reported higher values between 25% and 65% [161]. Likewise, other authors indicate that 80% of the bruises are located on the limbs [162]. Some factors, such as body weight or duration of transport, influence the prevalence of bruising [163]. In addition, small ruminants slaughtered directly from the farm displayed lower prevalence of bruises on carcasses [164]. Lameness is one of the important signs of the disease that compromises the welfare of animals, responsible for long-term pain and deterioration of normal behavior in small ruminants. Although the assessment of hoof health at the slaughterhouse can be reliable, scarce research is available [165,166]. Body condition score in small ruminants has been described as an important indicator of welfare on farms [167]. Although this parameter can be measured through the classification of the carcass [168], there is scarce research available studying the association between transport conditions (duration, distance, density, etc.), weight loss, and the quality of the meat [159,169].

## 4. Surveillance of Antibiotic Resistance in the Slaughterhouse

The use of antibiotics in veterinary medicine and animal production for the treatment, prevention, and control of diseases has resulted in healthier and more productive animals [170]. However, the continued, even excessive, use of antibiotics contributes to the emergence and spread of resistant bacteria.

In the European Union, the national veterinary authority of each member state must monitor antimicrobial resistances (AMR) of zoonotic and indicator bacteria in food-producing animals and derived meat as defined by law [171,172] and submitted to EFSA [9]. 

Member States must ensure that the surveillance system provides, at a minimum, relevant information on a representative number of strains of *Salmonella* spp., *Campylobacter jejuni,* and *Campylobacter coli* originating from cattle, pigs, and poultry as well as food of animal origin derived from these species. In addition, information must be provided for commensal indicator for *E. coli*, *Enterococcus faecalis*, and *Enteroccoccus faecium*. In addition, the monitoring of *Salmonella* spp. and *E. coli* producing extended-spectrum b-lactamases (ESBL), AmpC B-lactamases (AmpC), and carbapenenmases must also be monitored [172].

The antimicrobial resistance of zoonotic bacteria isolated from slaughterhouses is presented in Table 8. In slaughterhouses, antimicrobial resistance studies are carried out mainly in chicken and pigs. However, scarce research is available for cattle, sheep, and goats. According to the reviewed data, tetracycline displayed the highest antimicrobial resistance followed by some cephalosporins (cephalotim or cefpodoxime) and β-lactams (ampiciline or cefotaxime). However, these results must be carefully observed as antimicrobial resistance of most commonly used antimicrobials in large animal medicine (i.e., ceftioufur, tulatrmycine, gamitromicine, enrofloxacine, doxicicline, or florfenicol) has been scarcely studied at slaughterhouse level.

Although the data indicated that most bacteria are still sensitive to individual antimicrobials, the multi-resistance characteristics of foodborne pathogens have been referred to as an important issue [173,174]. 

In addition, the presence of antimicrobial resistance (AMR) genes in foodborne pathogens at the slaughterhouse level has also been studied. MRSA extended-spectrum ß-lactamase-producing *E. coli* has been detected in the CTX-M1 group in swine [173]. Other reports detected a high prevalence of extended-spectrum ß-lactamase producers (from *E. coli* and *K. pneumoniae*), carbapenemase producers (OXA-244-producing *E. coli*), colistin-resistant *E. coli* (MCR-1), and fosfomycin-resistant *E. coli* (fosA3, fosA4 and fosA6) [175]. Regarding *Listeria* spp., it has been reported that all samples isolated from pigs at the slaughterhouse had the AMR gene fosX indicating genotypic resistance to fosfomycin [176]. With regard to *Salmonella* spp., extended-spectrum β-lactamase (blaCTX-M-1, blaCTX-M-3, blaCTX-M-13, blaCTX-M-14, blaCTX-M-15, and blaSHV-12 (ESBL types); blaCMY-2 (AmpC type); and blaTEM-1 and blaOXA-1) and plasmid-mediated quinolone resistance (PMQR) (included qnrA, qnrB, qnrS, and aac(6′)-Ib-cr) were detected in Egypt [177]. Other studies indicate that environmental microflora in the slaughterhouse with AMR genes may also spread through the carcasses. Therefore, AMR surveillance programs should include carcasses and slaughterhouses surfaces and/or equipment as well as proper cleaning and disinfection programs [178,179].

## 5. Surveillance of Zoonotic Agents in the Slaughterhouse

Emerging and re-emerging infectious diseases continue to be a public health problem. In addition, a large part of the infectious diseases that affect people are zoonotic. [180]. Currently, many of the zoonotic diseases that affect livestock are controlled through national eradication programs, such as bovine tuberculosis or large and small ruminant brucellosis, while others seem to have gained some importance in recent years, such as Q fever. Currently, most of these zoonotic diseases are considered occupational diseases (i.e., workplace infections) associated with farmers, veterinarians, or slaughterhouse workers or as a result of a foodborne outbreak. Thus, the adoption of good manufacturing practices both in the slaughterhouse and in the food industry allows controlling most of them [181]. This topic discusses the importance of the slaughterhouse as a surveillance center for the different zoonotic agents described in the European Union One Health 2021 zoonoses report [182]. 

Bovine tuberculosis (BTb) and brucellosis of large and small ruminants are controlled through national eradication programmes. Currently, its prevalence is low and in many countries it has been eradicated [183]. From the point of view of public health, these zoonotic diseases are important in those countries where there are no eradication programs. Therefore, BTb represents one of the main causes of condemnation as previously described in Section 2.1, where its prevalence is highly variable (see Table 2) [43,50,56,57,58,59]. 

Regarding brucellosis, as described for BTb, there are prevalence studies in slaughterhouses in countries where there are no national eradication programs put in place. According to the reviewed data, the prevalence of brucellosis in the slaughterhouse varies between 3% and 15% [184,185,186,187,188].

Campylobacteriosis is the most commonly reported foodborne gastrointestinal infection in humans in the EU associated mainly with the consumption of chicken meat and raw milk [182]. According to the latest EFSA report on zoonosis, 25 out 65,895 samples of meat and meat products (including carcasses and fresh meat/ready-to-eat (RTE), cooked and fermented product) were positive. However, scarce information is available on non-poultry sources. The prevalence of *Campylobacter* spp. at slaughterhouse is approximately 20% (ranged from 5% to 40%) in cattle [189,190,191,192,193,194,195]. 

**Table 8 vetsci-10-00167-t008:** Antimicrobial resistance of several foodborne and zoonotic agents isolated from slaughterhouses.

Microorganism		CAM	CAM coli	CAM	CAM Jejuni	CPER Type a	CPER Type b	ARCO	ARCO	ECOL	ECOL STEC	ECOL	ECOL	ECOL O157	LISTERIA mono	LISTERIA mono	LISTERIA mono	LISTERIA mono	SALMO	SALMO	SALMO	NAS	S.AUREU
Species		C	S	S	S	C	Sh	C	Sh	C	C	S	V	V	V	S	-	S	S	S	S	S	S
Prevalence (%)		25.6	-	41.6	49.5	40	37.5	25	28.6	8.6	-	28.3	70	20.4	10.2	18.8	25.8	-	17.4	-	-	-	37.1
Sample		C	F	C–F	C–F	C	C	F	F	F	C	C	C	C	C	C	SE	C	C–F	F	F	C	C
Antimicro Family	Antimicro Type																						
Ansamy	Rifanpim	-	-	-	-	-	-	100	100	-	-	-	-	-	-	-	56	-	-	-	-	-	-
Aminoglucosides	Amikacin	-	-	-	-	-	-	-	-	-	-	-	-	-	18	-	-	-	-	-	-	-	-
	Gentamicin	2.6		38	25.9				11.1			7.9							16.2	14.2			
	kanamycin	-	-	-	-	-	-	-	-	26.3	-	-	-	-	18	-	-	-	15	-	-	-	-
	Neomicyn	-	-	-	-	-	-	-	-	-	-	-	-	-	-	-	-	-	12.5	-	-	-	-
	Spectino																			6.1			
	Strepto	24.3	68.9	90	76.9	-	-	-	-	-	22.2	-	-	18	18	-	-	-	28.7	-	54.7	-	-
	Tobramycin	-	-	-	-	-	-	-	-	-	-	-	-	-	30	-	-	-	15	-	-	-	-
Amphenic	Chloramphe	-	-	-	-	-	-	35	44	-	-	49.3	-	-	-	-	-	-	46.2	-	7.1	6.4	7.7
	Florfenicol	-	-	-	-	-	-	-	-	-	-	-	-	-	-	-	-	-	37.5	12.2	-	-	-
Beta lactams	Amox-Clav	-	-	-	-	-	-	-	-	-	-	-	100	-	-	43	-	-	27.5	-	-	-	-
	Ampicilin	-	-	-	80.9	-	-	-	-	100	-	-	100	5	-	-	-	14.3	75	32.7	35.71	-	-
	Benzylpen	-	-	-	-	-	-	-	-	-	-	-	-	-	-	-	100	-	-	-	-	-	-
	Cefepime	-	-	-	-	-	-	-	-	-	-	-	-	-	-	-	-	100	-	-	-	-	-
	Cefotaxime	-	-	-	-	-	-	-	-	100	-	-	-	-	100	-	-	-	-	-	-	29.8	-
	Cefoxitin	-	-	-	-	-	-	-	-	-	-	18.8	-	10	-	-	-	-	-	-	-	8.5	-
	Ceftazidime	-	-	-	-	-	-	-	-	-	-	-	-	-	-	-	-	85.7	-	-	-	-	-
	Mezlocilin	-	-	-	-	-	-	-	-	-	-	-	-	-	-	-	-	-	12,5	-	-	-	-
	Oxacillin	-	-	-	-	-	-	-	-	-	-	-	-	-	-	-	92	-	-	-	-	83.0	-
	Penicillin	-	-	-	-	-	-	-	-	-	-	-	-	-	-	-	-	-	-	-	-	34	-
Carbapen	Imipenem	-	-	-	-	-	-	-	-	-	-	-	-	-	-	-	100	-	-	-	-	-	-
Cephalos	Cefazolin	-	-	-	-	-	-	95	100	-	-	-	-	-	-	-	-	-	-	-	-	-	-
	cephalothin	-	-	-	-	-	-	100	100	-	-	-	-	5	18	-	-	-	-	-	-	-	-
	cefpodoxime	-	-	-	-	-	-	-	-	100	-	-	-	-	-	-	-	-	-	-	-	-	-
	Ceftriaxone	-	-	-	-	-	-	100	100	-	-	-	-	-	-	-	-	-	-	-	-	-	-
Fusidane	Fusicid acid	-	-	-	-	-	-	-	-	-	-	-	-	-	-	-	100	-	-	-	-	-	7.7
Glycopep	Vancomycin	-	-	-	-	-	-	100	100	-	-	98.1	-	-	-	-	-	14.3	-	-	-	-	-
Lincosam	Clindamycin	-	-	-	-	-	-	-	-	-	-	-	-	-	-	100	-	-	-	-	-	-	100
	Lincomycin	-	-	-	-	-	-	-	-	-	-	-	-	-	-	100	-	-	-	-	-	-	-
	Pirlimycin	-	-	-	-	-	-	-	-	-	-	-	-	-	-	100	-	-	-	-	-	-	-
Macrolides	Erythrom	24.3	25.2	48	56.1	-	-	-	-	52.6	-	-	-	-	18	-	-	28.5	-	-	-	4.3	30.8
Polimixins	Colistin	-	-	-	6.8	-	-	-	-	-	-	-	-	-	-	-	-	-	-	-	-	-	-
Quinolon	Ciprofloxa	38.8	34.7	70	95	-	-	-	-	-	-	39.9	25	-	-	-	4	100	10	2.04	-	2.1	92.3
	Enrofloxacin	-	-	-	-	-	-	-	-	7.6	-	-	-	-	-	-	-	-	13.7	-	-	-	-
	Marbofloxa	-	-	-	-	-	-	-	-	-	-	-	-	-	-	14	-	-	-	-	-	-	-
	Nalidixic ac	38.8	35.3	75	87.7	-	-	46.1	55.5	-	-	-	-	25	100	-	-	-	-	10.2	-	-	-
Reference		[194]	[195]	[196]	[174]	[197]	[197]	[198]	[188]	[199]	[200]	[201]	[202]	[203]	[204]	[176]	[204]	[205]	[206]	[207]	[208]	[209]	[210]
Specie		C	S	S	S	C	Sh	C	Sh	C	C	S	V	V	V	S	-	S	S	S	S	S	S
Prevalence (%)		25.6	-	41.6	49.5	40	37.5	25	28.6	8.6	-	28.3	70	20.4	10.2	18.8	25.8	-	17.4	-	-	-	48.6
Sample		C	F	C–F	C–F	C	C	F	F	F	C	C	C	C	C	C	SE	C	C–F	F	F	C	C
Antimicro family	Antimicro type																						
Sulfonam	Trime-sulfa	-	-	-	-	-	-	100	100	73.7	-	68.8	-	18	-	-	48	-	30	10.2	4.76	4.3	100
Tetracyclin	Doxicicline	-	-	-	-	-	-	-	-	10.5	-	-	-	-	-	79	-	-	-	-	-	-	-
	Tetracicline	20.9	64.9	90	97.2	45.8	92.3	-	-	-	-	90.7	62.5	5	-	-	44	100	81.2	89.8	66.6	21.3	100
Reference		[194]	[195]	[196]	[174]	[197]	[197]	[198]	[188]	[199]	[200]	[201]	[202]	[203]	[204]	[176]	[204]	[205]	[206]	[207]	[208]	[209]	[210]

Antimicro: antimicrobial; Ansamy: ansamycin; Amphenic: amphenicols; Chloramphe: chloramphenicol; Amox-Clav: amoxicyllin and clavulamin acid; Benzylpen: benzylpenicillin; Spectino: spectinomycin; Strepto: streptomicyn; Carbapen: carbapenem; Cephalos: cephalosporins; Glycopep: glycopeptides; Lincosam: lincosamides; Erythrom: erythromicin; Ciprofloxa: ciprofloxacin; Marbofloxa: marbofloxacin; Nalidixic ac: nalidixic acid; Sulfonam: sulfonamides; Trime-sulfa: trimethoprim-sulfamethoxazole; Tetracyclin: tetracyclines; Quinolon: quinolones. NAS:Non-aureus staphylococci; Specie: C = cattle, S = swine, Sh = sheep; Sample: C= carcass, F: feces, SE = slaughter environment.

However, prevalence values above 90% have been reported in sheep [211] and calves [192]. Although prevalence of campylobacter in cattle is low, the highest prevalence values in calves and sheep suggest that preventive measures should be applied on the farm. 

Chlamydiasis or enzootic abortion, caused by *Chlamydia abortus*, is responsible for abortion in the last 2–3 weeks of pregnancy in sheep. It can also affect other farm animals such as goats, cattle and pigs with little importance. Most studies on the prevalence of *C. abortus* are serological surveys conducted on farms. At the slaughterhouse, the prevalence of *C. abortus* in uterine samples was about 30% [212] in non-pregnant ewes and about 45% in pregnant ewes [213]. Seroprevalence surveys at slaughterhouse indicate prevalence values about 11% in sheep and goats [214]. Data on zoonotic *Chlamydia abortus* are not presented in the latest zoonotic EFSA report [182]. In pigs, *Chlamydia suis* has suggested as a public health treat since it is phylogenetically closely related to the human pathogen *Chlamydia trachomatis*. Research on the prevalence in slaughterhouse is scarce although some authors related rates of 45%–90% in faucal samples [215] and approximately 40% in conjunctival samples [216] Although its zoonotic potential is currently unclear, conjunctivitis caused by this bacterium or together with *C. trachomantis* has been observed in trachoma endemic regions [217].

Regarding *E. coli*, some strains produce toxins, called verotoxins or shiga like toxins that can cause serious illness in people. These strains are called by different names: *E. coli* verotoxigenic (VTEC), shiga-toxin producing *E. coli* (STEC), or *E. coli* enterohemorrhagic (EHEC). The main serotypes of this group are: O157:H7, O104:H4, O26, O103, O111 and O145. Ruminants are considered the main reservoir of STEC, and cattle are the main contributor to the disease in people. The prevalence of STEC in cattle carcasses ranged from 1.5% to 15% [218,219,220,221,222]. However, higher prevalence STEC values up to 49% have been reported [223]. With regards of non-O157 [221], a prevalence of 3.2% was reported in cattle carcasses (including O26, O45, O103, O111, O121, and O145). However, other studies report higher prevalence values around 20% (O166), 16% (O146), 13% (O44), 32% (O111), and 19% (O26) [224]. Cattle younger than 24 months old showed STEC values seven times higher than cattle older than 24 months old [219]. Another report observed a STEC reduction of about 0.5 CFU/cm^2^ in the pre-chill stage compared to the pre-evisceration stage in beef slaughterhouse [220].

In small ruminants, the prevalence of STEC ranged from 1.25% to 4.6% [219,222] while the prevalence of VTEC is about 14% in sheep carcasses [225]. Although ruminants are the main carriers of STEC, prevalence from 1.8% to 4.6% in pig carcasses has been described in the literature [226,227,228,229].

Although the prevalence of STEC in cattle carcasses is low, it was the fourth most commonly reported foodborne gastrointestinal infection in humans in the EU [182,212]. Since STEC represents a public health hazard, some carcass decontamination techniques have been suggested, such as the application of organic acids, inorganic acid, or bacteriophages, among others [230]. In addition, the importance of its control is not only associated with its role as a foodborne pathogen, but also with its role in the transmission of antimicrobial resistance as previously discussed.

*Cryptosporidium* spp. and *Giardia* spp. are ubiquitous protozoan parasites that infect a broad range of animals and humans. Both parasites are of medical and veterinary importance. Studies on prevalence of both parasites are scarce so the real threat to public health could be underestimated [231]. The prevalence of *Cryptosporidium* spp in cattle was 33.8% in samples taken from farms and slaughterhouses [232]. In contrast, other studies reported lower prevalence values ranging from 8.3% to 16.7% and from 0% to 16.7% for *Cryptosporidium* spp. and *Giardia* spp. in samples of slaughterhouse effluents, respectively [233].

In pork, Cryptosporidium oocysts were identified microscopically in 29% in finisher pigs and 10% of sows [234]. Although three outbreaks and 34 cases were reported in the EU in 2020 EU [182], the highest prevalence of both parasites reported in farm animals [234] (especially in calves and heifers than adult cattle) could be a health risk for farmers, veterinarians, and slaughterhouses staff through direct fecal–oral transmission.

Q fever or coxiellosis is a bacterial disease caused by *Coxiella burnetii* distributed worldwide, with special importance in ruminants, causing abortions and perinatal deaths. Q fever is generally an occupational zoonosis. Although Q fever affects a large number of animal species, both domestic and wild, livestock is considered to be the main source of infection for humans [235]. Q fever in humans is a notifiable disease at EU level. On the other hand, there is no harmonized control system in animals. Therefore, most diagnoses are based on clinical investigations and passive monitoring, such as detection of reproductive diseases in bulk tank milk. Most Q fever seroprevalence studies are conducted by blood sampling on farms. At the slaughterhouse, serological surveys reported prevalence in cattle about 24% in Brazil, which 12% displayed positive qPCR results suggestive of active or recent infection [236]. However, lower prevalence has also been reported, ranging from 1.5% to 9.4% [237,238]. The prevalence of Q fever in sheep is lower than in cattle. Serological surveys conducted in slaughterhouses yielded prevalence values ranging between 4.3% to 16.5% [237,239] by ELISA. Likewise, only 12.7% of positive sera were positive by PCR [237]. In goats, the prevalence of Q fever seems to be higher than in cattle and sheep. Thus, values of 22%–28.5% have been reported in serological surveys in slaughterhouses [239,240]. Although the seroprevalence, mainly in cattle and goats, is moderate, its impact on public health is low. The epidemiological pattern of Q fever is similar to that of other intracellular infectious diseases (e.g., tuberculosis) in such a way that the infection does not always derive from contact with the microorganism and, on the other hand, not all cases of infection cause clinical disease. Q fever presents in an acute or chronic form according to the characteristics of the host. Surveillance of Q fever trough the slaughterhouse allows it to be monitored in different geographical locations depending on the origin of the animals. These data can help understand seasonal variations, as well as the study of anthropozonotic outbreaks [235].

Leptospirosis is a bacterial zoonosis of economic importance in the livestock industry because it causes abortions, stillbirths, infertility, and decreased milk production. The disease is caused by pathogenic spirochetes of the genus Leptospira. Surveillance for leptospirosis in slaughtered animals is scarce and no positive results were obtained according to the latest EFSA report [182]. Leptospirosis showed low rates in the EU. Transmission occurs primarily through contact with urine from infected animal reservoirs, through penetration of mucous membranes or broken skin, with rodents being the most important source for human and animal infections. Since the risk factors include farming, exposure to wildlife and rodents, the study of the prevalence of *Leptopira* spp. in the slaughterhouse can be used as an environmental monitoring system for zoonotic outbreaks. 

Serological surveys in cattle at slaughter vary from 3% to 80% [241,242,243,244,245,246] among studies. Thus, seroprevalence of around 80% in cattle in which 20% of them were PCR positive in kidney samples [245]. Similar results were also observed by other authors [247] with a seroprevalence of about 73%, although only 40% of them showed positive PCR results in urine and kidney samples. Furthermore, kidneys with white spot lesions are five times more likely to be infected with pathogenic Leptospira species than other lesions [246]. In addition, the prevalence not only varies between regions and season, but also according to the diagnostic test. Thus, other research indicated that 78% of bovine kidneys sampled were positive by direct fluorescent antibody staining while 29.7% and 8.1% were positive by PCR and culture, respectively [247].

In small ruminants, some authors [245,248,249,250] reported values of seroprevalence in the slaughterhouse of 11%, 18% and 40% respectively. 

Although lower seroprevalence values (ranging from 2% to 25%) have been published for goats [245,249]. In pigs, a seroprevalence of 32% has been reported at the slaughterhouse [251].

Surveillance data for *Leptospira* spp. at the slaughterhouse showed large variation between species, location season and diagnostic technique. However, the data highlight that leptospiral infection is asymptomatic in most livestock. Thus, implementation of control measures, such as the prevention of rodents on the farm and/or the vaccination of livestock, may reduce the prevalence in animals and further decrease public health problems with special relevance in risk groups (veterinarians, farmers, and slaughterhouse workers).

Zoonotic taeniasis is caused by the adult stage of *Taenia solium*, *Taenia saginata*, or *Taenia asiatica* which are considered neglected tropical diseases by the World Health Organization. The life cycle of these three species is very similar and includes an intermediate host: pigs in the case of *T. solium* and *T. asiatica* and cattle in the case of *T. saginata*. By eating meat (pork/*T. solium*, *T. asiatica*; beef/*T. saginata*) containing live cysticerci, humans develop taeniasis, which is practically asymptomatic but is the main risk factor for intermediate hosts. *T. saginata* causes bovine cysticercosis, while *T. solium* and *T. asiatica* cause swine cysticercosis, which is of veterinary and economic importance. *T. solium* cysticerci causes neurological disease in humans: neurocysticercosis [252,253]. 

According to EFSA data, surveillance in slaughterhouses shows lower prevalence values in both farm and game animals. However, higher values have been suggested [254] due to the low sensitivity of traditional meat inspection. Therefore, the seroprevalence of 0.26% in Belgian cattle at slaughterhouse, 10-fold higher than the prevalence reported by veterinary authorities, indicates that classical meat inspection techniques detect only a minor fraction [255]. In addition, other authors [256] reported a sensitivity of post-mortem inspection, compared to a gold standard of stereoscopic microscopy, of 52.4%.

The prevalence of bovine cysticercosis varies geographically. In developed countries, surveillance in slaughterhouses ranged from 0.10% to 0.25% [3,254,257,258]. However, prevalence values from 7% to 27% have been reported in developing countries [259,260,261] probably associated with the lack of antiparasitic treatments. Regarding cysticercosis in pigs, lower prevalence values were observed in developed countries than in developing countries [262,263,264,265]. However, the absence of cyst in the carcass has also been reported after meat inspection [263]. In a six-year surveillance program for pig lesions from slaughterhouses, only two parasitic muscle granulomas, both PCR-negative, were submitted for laboratory diagnostic [265]. According to previous data, the prevalence of cysticercosis in the slaughterhouse is low. However, classical meat inspection should be considered together with molecular diagnosis. Although zoonotic cysticercosis outbreaks in developed countries are low, a proper risk assessment must be performed for visual only meat inspection of pig carcasses.

Hydatidosis/echinococcosis is a zoonotic parasitic disease caused by tapeworms of the genus Echinococcus. Humans become infected through ingestion of parasite eggs in contaminated food, water, soil, or through direct contact with animal hosts. *E. multilocularis* caused alveolar echinococcosis while *E. granulosus sensu lato* caused cystic echinococcosis. At the slaughterhouse level, *E. granulosus s. l.* revealed importance for public health since it presents a pastoral cycle in which the main livestock species (sheep, cattle, goats and pigs) represent its intermediate host. Most of the prevalence studies at the slaughterhouse level are carried out in developing countries with a wide range of prevalence as indicated in Table 9. 

According to EFSA data [182], surveillance of livestock in slaughterhouse showed a prevalence about 0.21% and 0.96% in cattle and small ruminants, respectively, in 2020. Prevalence studies of echinococcosis in slaughterhouse in Europe are scarce. In France, a survey in slaughterhouse revealed a prevalence of 0.002% in sheep and 0.00001% in cattle. Moreover, one pig was positive to *E. granulosus s. l.* Neither goats nor horses presented echinococcus cyst [281]. In Italy, a prevalence of 62.9% and 28.0% in sheep and goats have recently been reported [275]. In Greece, a prevalence of hydatidosis of 31.3% was reported, with 8.7%, 4.8%, and 1.7% in sheep, goats, cattle, and pigs, respectively, at slaughterhouse [274]. With regard to pigs, a higher prevalence of hydatidosis was observed in Switzerland in breeding pigs (0.11%) than in fattening pigs (0.008%) [284]. 

In Hungary, the reported infection rate was 0.013% in sheep, 0.007% in cattle, and 0.001% in pigs [276]. Although the lower prevalence is in accordance with the EFSA data [182], given the fact that the hydatidosis-positive animals in the study come from family farms, farmers should be encouraged to implement appropriate deworming programs for definitive hosts, as well as proper cooking of meat. In addition, in other developed countries such as Australia, the prevalence in cattle at the slaughterhouse seems to be higher (see Table 9). Although there is no treatment for echinococcosis in livestock, the risk of zoonotic hydatid disease from direct contact with infected animals or from contaminated food appears to be low in the EU. However, domestic slaughter of cattle is allowed in some European countries (under specific conditions) which highlights this public health issue. 

Preventive measures to avoid zoonotic infection by *Echinococcus* spp. include deworming of dogs, slaughterhouse hygiene, controlling stray dog populations, restricting home slaughter of sheep and other livestock, not consuming any food or water that may have been contaminated with dog fecal matter, washing hands with warm water, proper cooking, and public health education.

For this reason, proper cooking or freezing of the meat is essential to avoid parasitic-borne infections although it is best for the farmer to contact a veterinarian to carry out an adequate inspection of the meat. Positive cases of hydatidosis detected in the slaughterhouse should be cause for public health concern. Therefore, OVI should advise farmers to implement corrective measures on the farm, such as proper deworming of farm dogs or avoiding contact between livestock and wild carnivores thought appropriate biosecurity measures. The risk of foodborne salmonellosis is low since the meat of livestock species is usually eaten cooked. However, foodborne outbreaks can occur through cross-contamination between meat and ready-to-eat foods or due to improper cooking of meat (low temperatures). Regarding salmonella, control in bovine, sheep, goats, and swine carcasses is mandatory after dressing but before chilling to assess the hygiene process of the slaughterhouse [12]. The sampling plan is N = 50 (number of samples) and C=2, where a negative result is the absence in the area tested per carcass. The 50 samples are derived from 10 consecutive sampling sessions according to the rules and sampling frequencies established in the Regulation. The “C” value represents the maximum number of samples where the presence of salmonella is detected [12]. Positive results indicate a poor hygiene process. Regarding the prevalence of *Salmonella* spp. in carcasses analyzed by the competent authorities of different EU member states, it was 1.6% for cattle, 0.45% for sheep, 1.2% for goats, and 3.6% for pigs [182]. For *L. monocytogenes* and *S. aureus*, non-mandatory analyses are required for carcasses. Research on the prevalence of *L. monocytogenes* and *S. aureus* in carcasses is presented in Table 10. Data on the prevalence of both microorganisms on carcasses are scarce. In general, the prevalence values of both microorganisms vary greatly between species and countries. Furthermore, most of the studies on *S. aureus* were aimed at investigating the presence of MRSA of animal origin. In the case of unsatisfactory results, improvements in the slaughter hygiene and review of meat processes must be applied. 

## 6. Conclusions

Food safety and quality are the main demands of consumers. Likewise, clear, truthful, and direct information about food, based on science, is essential to build trust among consumers and advance food safety.

From the point of public health, the objective of the slaughterhouse is to guarantee the safety of meat from the perspective of animal pathology and disease, where meat inspection represents an essential tool to control animal diseases and guarantee public health. 

In addition, the slaughterhouse can, and should, be used as a central point in the surveillance of other animals and other current issues, such as the epidemiology of livestock diseases (including those zoonotic), the control of animal welfare in the farm, surveillance of zoonotic agents responsible for food poisoning, as well as the monitoring and control of antimicrobial resistance.

The advantage of the slaughterhouse as a surveillance center gives us the possibility of controlling all these aspects in a single place, with a large animal sample collection and a wide geographical area. However, we must not consider these controls and/or monitoring programs in the slaughterhouse as the last defensive barrier for animal health and public health, but rather as complementary surveillance schemes to those carried out on farms. In recent years, the modernization of meat inspection has been discussed, moving from a traditional inspection with fixed incisions and palpations, to inspection based on risk analysis, taking into account evolving scientific knowledge, risks, and hazards (as has been proposed for post-mortem inspection of pigs and poultry), to a visual inspection. This modernization is due to aspects, such as the increase in the slaughter rate, improvements in animal protection and welfare, and the improvement in the health status of livestock. This modernization of the post-mortem meat inspection tries to prioritize the hazards since many of them are inapparent during the classic meat inspection. However, the differences between species and types of production make the classical meat inspection a fundamental tool to guarantee public health.

Through the different types of lesions observed during ante-mortem and post-mortem inspections, the slaughterhouse continues to be a fundamental tool for the knowledge and control of diseases in primary production. Studies related to diseases and/or lesions found during meat inspection reveal that the existing data are scarce and outdated, not taking advantage of their potential for disease control. In addition, the reviewed research is carried out in developing countries where aspects, such as the implementation of prophylaxis programs, farm biosecurity, animal identification, and/or traceability systems and disease eradication programs, are scarce or nor-existent. This reveals an underestimation of the meat inspection data. In addition, another limitation observed in this review is the absence of information about the existence of national databases of total and partial condemnations. This information is not only fundamental from the point of view of animal and public health (animal and human), but also economically important in the implementation of preventive programs.

Animal welfare is an aspect that is increasingly demanded by consumers. In the slaughterhouse, ante-mortem and post-mortem inspection provide valuable individual data on animal welfare, not only in primary production, but also during transport and slaughter. Although a variety of indicators have been proposed for the evaluation of animal welfare, their observation and/or quantification is not always possible in the slaughterhouse. Aspects, such as body condition, presence of lesions and/or lameness or behavior in the pens, in the ante-mortem inspection, as well as lesions in the organs and/or carcasses, bruises in carcasses, or hoof health in the post-mortem inspection, allow us to assess the welfare of livestock at the slaughterhouse. In addition, data on animal movement and transport contribute to assessing the welfare of livestock. 

The use of antibiotics in animal production has allowed not only the treatment, but also the prevention of diseases in livestock, increasing their productivity and well-being. The problem derived from its continuous and/or incorrect use has led to an important public health issue, antimicrobial resistance. Although in the EU there are policies and surveillance programs regarding the resistance of certain zoonotic agents (*Salmonella* spp., *Campylobacter* spp., *E. coli*, or Enterococcus), given the variety of zoonotic agents and antibiotics, the investigation of antimicrobial resistance in livestock intended for consumption is scarce, mainly in cattle, sheep, and goats. As discussed above, most zoonotic agents are sensitive to the antibiotics studied.

However, it is important to highlight that there are few studies that evaluate the antimicrobial resistance of the main zoonotic agents to the main antimicrobials used in veterinary and human medicine.

The slaughterhouse represents a fundamental control point in the surveillance and control of zoonotic agents responsible for foodborne infections and intoxications. Agents, such as *Salmonella* spp., *E. coli*, or *L. monocytogenes*, are mandatorily controlled. Although the data published in the latest EFSA report indicate a low prevalence of these agents in slaughterhouses across Europe, aspects, such as the lack of harmonization in terms of control and communication, may underestimate their real prevalence. In addition, it has been observed that the prevalence of zoonotic agents is highly variable in those countries where there is no existence of a specific policy for their control, programs for the eradication of zoonotic diseases, or implementation of prophylaxis programs in primary production.

Finally, the use of the slaughterhouse as surveilling center as proposed in this review requires human and financial resources. The OVI’s workload is currently high as they are not only responsible for the ante-mortem and post-mortem inspection, but also responsible for the control of animal documentation, the welfare of slaughter operations, and control of by-products.

The data collection regarding total and partial condemnations requires an OVI or an assistant in each slaughter line. For welfare monitoring, data on bruises and hoof/claw health seem to be two indicators that can be evaluated at the slaughterhouse, even those with high slaughter rates. However, this task requires at least one OVI exclusively for each one of these tasks. For the surveillance of AMR and/or control of zoonotic agents, apart from human resources, it is necessary to allocate a significant budget (including infrastructure in those places where it does not exist).

However, as previously discussed, the slaughterhouse cannot be considered as the last defense line of animal and public health, so these proposals for surveillance programs must be complementary to the control programs implemented in primary production.

Last but not least, the main challenge is to translate these surveillance plans into standardized monitoring programs adapted for each species and developed for primary production.

## Figures and Tables

**Table 1 vetsci-10-00167-t001:** Prevalence of cattle carcass condemnation.

Cattle Simple Size	Prev. Cond (%)	Place	Period	Reference
3042	0.40	Palestine	July–December 2018	[52]
1,162,410	0.59	Canada	2001–2007	[53]
Not indicated	0.5–1.00	US	2005–2011	[47]
58,483	0.25	Italy	2010–2016	[54]
1,439,868	0.70	France	2006–2010	[46]
Not indicated	0.29	UK	1969–1975	[55]
2741	0.21	Iran	2011–2012	[56]
42,434	0.20	Tanzania	1987–1989	[57]
115,186	0.11	Tanzania	2001–2007	[58]
37,717	0.51	Zambia	2000–2003	[50]
23,064	0.30	Ethiopia	2008–2012	[59]
85,980	0.05	Tanzania	2010–2012	[60]
4,000,372	4.60	CR	1995–2002	[61]
3,816,119	0.31	Switzerland	2007–2012	[42]
26,694,317	0.20	US	2003–2007	[62]
22,872	0.32	Turkey	June 2012–December 2012	[43]

% Prev. Cond.: prevalence condemnation; UK: United Kingdom, CR: Czech Republic; US: United States.

**Table 7 vetsci-10-00167-t007:** Organ/partial condemnations in pigs.

Organ/Part	%	Organ/Part	%	Country	Reference
Head	2.51	Lungs	31.53	Brazil	[99]
Heart	8.49	Spleen	3.82		
Intestine	6.00	Stomach	1.11		
Kidneys	14.44	Tongue	2.90		
Liver	15.24				

**Table 9 vetsci-10-00167-t009:** Hydatidosis prevalence at slaughterhouse.

Specie	Prev. (%)	Country	Reference
Cattle	59.30	Moldova	[266]
	46.90	Ethiopia	[267]
	20.50	Ethiopia	[268]
	17.90	Ethiopia	[269]
	16.00	Ethiopia	[260]
	15.20	Ethiopia	[270]
	9.40	Iran	[271]
	8.80	Australia	[272]
	5.30	Kenya	[273]
	4.80	Greece	[274]
	3.90	Mozambique	[275]
	0.07	Hungary	[276]
Sheep	62.90	Italy	[277]
	62.90	Lebanon	[278]
	61.90	Moldova	[266]
	31.30	Greece	[274]
	29.30	Ethiopia	[267]
	12.12	Iran	[277]
	7.63	Morocco	[279]
	0.77	Egypt	[280]
	0.10	Kenya	[273]
	0.01	Hungary	[276]
Goat	28.00	Italy	[277]
	20.90	Lebanon	[278]
	19.07	Iran	[281]
	10.70	India	[282]
	8.70	Greece	[274]
	6.70	Ethiopia	[267]
	2.00	Kenya	[273]
Pig	2.90	Mozambique	[275]
	1.70	Greece	[274]
	1.00	France ^1^	[283]
	0.24	Egypt	[280]
	0.01	Hungary	[276]

Prev.: prevalence; ^1^ only in Corsica island.

**Table 10 vetsci-10-00167-t010:** Prevalence of *L. monocytogenes* and *S. aureus* in carcasses.

Microorganism	Prev. (%)	Specie	Country	Reference
*L. monocytogenes*	43.00	Cattle	Belgium	[285]
	46.00	Cattle	Belgium	[286]
	15.50	Cattle	China	[287]
	14.00	Cattle	Ireland	[288]
	6.80	Cattle	Turkey	[289]
	6.00	Cattle	Brazil	[290]
	5.17	Cattle	Korea	[291]
	3.40	Cattle	Turkey	[292]
	2.50	Sheep	Turkey	[292]
	22.00	Pig	Belgium	[286]
	1.33	Pig	Spain	[293]
*S. aureus*	32.50	Cattle	South Africa	[294]
	5.70	Cattle	Greek	[295]
	4.50	Cattle	Greek	[296]
	12.50	Goat	Greek	[296]
	15.50	Pig	Greek	[295]
	8.00	Pig	Greek	[296]
	2.33	Pig	Nigeria	[297]
	7.00	Sheep	Greek	[298]
	5.00	Srum	Greek	[298]

Prev.: prevalence; Srum: small ruminants.

## Data Availability

Not applicable.

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
