# Peer review of "The Importance of the Slaughterhouse in Surveilling Animal and Public Health: A Systematic Review"

_vetsci, 2023, doi:10.3390/vetsci10020167_

Round 1
Reviewer 1 Report
Veterinary Sciences (ISSN 2306-7381)
Manuscript ID. vetsci-2161620
Type. Review
Title: The importance of the slaughterhouse as a surveillance centre for....
Authors: Juan García-Díez, et al.
The author(s) editorialized on the role of the slaughterhouse in detail in this manuscript which as the essential tool to control animal diseases and guarantee the public health. Traditionally, the role of the slaughterhouse is guaranteeing the safety of meat from the perspective of animal pathology and disease. However, it can be used for monitoring other aspects that influence not only the animal health but also the food safety and public health such as animal welfare, antimicrobial resistance or prevalence of foodborne and zoonotic diseases.
Overall, this manuscript is reasonably well written. However, several concerning points need to be addressed.
The authors described that traditional role of the slaughterhouse is guaranteeing the safety of meat from the perspective of animal pathology and disease. In addition, it can be used for monitoring other aspects that influence not only the animal health but also the food safety and public health such as animal welfare, antimicrobial resistance or prevalence of foodborne and zoonotic diseases.
On the other hand, the authors described that regarding control of diseases in livestock, scientific research is scarce and outdated, not taking advantage of their potential for disease control. Prevalence at slaughterhouse of zoonotic and foodborne agents seems to be low but lack of harmonization in their control and communication may underestimate its real prevalence. Most researches are carried out in developing countries where aspects such as implementation of prophylaxis programs, farm biosecurity, animal identification and/or traceability systems and disease eradication programs are scarce or event absent. The Information of antimicrobial resistances at slaughterhouse is scarce, mainly in cattle, sheep and goats. It is important to highlight that there are few studies that evaluate the antimicrobial resistance of the main zoonotic agents to the main antimicrobials used in veterinary and human medicine. The slaughterhouse represents a fundamental control point in the surveillance and control of zoonotic agents responsible for foodborne infections and intoxications.
The reviewer points out that the research of slaughterhouse for various foodborne and AMR pathogens possessed by bovine and swine, and condemnation rates of carcass or organs are implemented not only in developing countries but also in the countries with prophylaxis programs, farm biosecurity, animal identification and/or traceability systems and disease eradication programs around the world. Also, is it conceivable that even within the same country, different slaughterhouses may have different implementation conditions? In addition, Due to the circumstances of the government and slaughterhouse operators, there are limits to the amount of money, manpower, and time that can be spent on various responses at slaughterhouses. The authors should describe minimum level to be investigated at slaughterhouse.
Tips
1 It is unclear how the numbers (%) shown in the text correspond to the numbers (%) shown in each table.
2 The authors should check out the manuscript throughout because of so many typographical errors.
Author Response
Prof. Dr. Patrick Butaye
Editor-in-Chief
Veterinary Science
- “The importance of the slaughterhouse in surveilling animal and public health: a systematic review” by García-Díez et al.
On behalf of all the authors, I am grateful to the editor and reviewers for the attentive and detailed remarks which helped to improve the manuscript.
We send enclosed the revised version of the manuscript “The importance of the slaughterhouse in surveilling animal and public health: a systematic review” by García-Díez et al. (vetsci-2161620) with the comments incorporated in the text.
We have modified, corrected and revised carefully the manuscript according to the Reviewers’ comments, and detailed corrections are listed below point by point. The changes we have made are indicated by tracking changes.
We hope modifications introduced in the manuscript are clear and concise enough as required by the Editor in order to enable the manuscript as suitable for review and further publication. Responses to the comments and suggestions required by reviewers can be found below.
Yours sincerely,
Juan García Diez, DVM, MSc, PhD.
Animal and Veterinary Research Center. University of Trás-os-Montes e Alto Douro, 5001-801. Vila Real, Portugal.
Tel.: +351259350659, Fax: +351259350480
e-mail: juangarciadiez@gmail.com
Reviewer 1´s comments
The suggestions of this reviewer were correctly addressed
The author(s) editorialized on the role of the slaughterhouse in detail in this manuscript which as the essential tool to control animal diseases and guarantee the public health. Traditionally, the role of the slaughterhouse is guaranteeing the safety of meat from the perspective of animal pathology and disease. However, it can be used for monitoring other aspects that influence not only the animal health but also the food safety and public health such as animal welfare, antimicrobial resistance or prevalence of foodborne and zoonotic diseases. Overall, this manuscript is reasonably well written. However, several concerning points need to be addressed.
The authors described that traditional role of the slaughterhouse is guaranteeing the safety of meat from the perspective of animal pathology and disease. In addition, it can be used for monitoring other aspects that influence not only the animal health but also the food safety and public health such as animal welfare, antimicrobial resistance or prevalence of foodborne and zoonotic diseases.
On the other hand, the authors described that regarding control of diseases in livestock, scientific research is scarce and outdated, not taking advantage of their potential for disease control. Prevalence at slaughterhouse of zoonotic and foodborne agents seems to be low but lack of harmonization in their control and communication may underestimate its real prevalence. Most researches are carried out in developing countries where aspects such as implementation of prophylaxis programs, farm biosecurity, animal identification and/or traceability systems and disease eradication programs are scarce or event absent. The Information of antimicrobial resistances at slaughterhouse is scarce, mainly in cattle, sheep and goats. It is important to highlight that there are few studies that evaluate the antimicrobial resistance of the main zoonotic agents to the main antimicrobials used in veterinary and human medicine. The slaughterhouse represents a fundamental control point in the surveillance and control of zoonotic agents responsible for foodborne infections and intoxications.
The reviewer points out that the research of slaughterhouse for various foodborne and AMR pathogens possessed by bovine and swine, and condemnation rates of carcass or organs are implemented not only in developing countries but also in the countries with prophylaxis programs, farm biosecurity, animal identification and/or traceability systems and disease eradication programs around the world.
- 1. Also, is it conceivable that even within the same country, different slaughterhouses may have different implementation conditions? In addition, Due to the circumstances of the government and slaughterhouse operators, there are limits to the amount of money, manpower, and time that can be spent on various responses at slaughterhouses. The authors should describe minimum level to be investigated at slaughterhouse.
RE: questions regarding manpower and economical allocation were added in the conclusion.
- 2. It is unclear how the numbers (%) shown in the text correspond to the numbers (%) shown in each table.
RE: information was added to the text.
- 3. The authors should check out the manuscript throughout because of so many typographical errors.
RE: the entire manuscript has been revised and improved.
REVIEWER 2´s Comments
- 1. Additionally, it would be very interesting and relevant to know more about any reported practices around disposal of, or interaction with, condemned carcasses and organs. This has huge relevance to human, livestock and wildlife health too.
RE: disposal by-products of slaughterhouses are not described in text since in Europe, all by product are destroyed by incineration. Classification, traceability, transportation, and destruction are defined by law, specifically by Regulation (EC) No 1069/2009 of the European Parliament and the council of 21 October 2009 laying down health rules as regards animal by-products and derived products not intended for human consumption and repealing Regulation (EC) No 1774/2002 (Animal by-products Regulation. Since in other countries (mainly in developing countries) burial and landfilling is a common practice, it may imply certainly risk for animal and public health. However, authors have not included information about these topics as well other topics such as wastewater management or health risk of slaughter staff. The main objective of the review is highlighting the potential of slaughterhouse as surveillance point of several aspects (e. g. AMR, welfare, etc.) that can improve both animal and human health in cost-effective way.
- 2. Consider simplifying the title to: “The importance of the slaughterhouse in surveilling animal and public health: a systematic review”
RE: the title was simplified as suggested.
- 3. Conceptual comment to the summary and abstract: perhaps this is my North American perspective coming through but while I do see the perspective that slaughterhouses can serve as important monitoring hubs, I think the ‘role’ of guaranteeing the safety of meat as described by the authors, and as they allude to themselves, starts well before the animal reaches the point of slaughter. One could say slaughterhouses and practices are a last line of defense, but I would not ascribe the full role there. Unless the authors mean because of the safety checks in place at slaughterhouses? Because if that is what they mean then some clarification is required on their part.
RE: We fully agree with the reviewer's comments. Animal health, food safety and public health must be guaranteed at all stages of the food chain, from “farm to fork”. We intend to show, in this review, that the slaughterhouse, apart from guaranteeing the safety of meat through meat inspection, can be used for many other studies and/or controls such as monitoring animal health, welfare or antibiotic resistance. Since the slaughterhouse is a “meeting point” for animals from different geographical origins, allows us to centralize this control/study/monitoring. In accordance with the reviewer, this control/study/monitoring in the slaughterhouse should not be considered as a last line of defense but rather as a complement to the different controls carried out at farms (primary production). This information has been added in the abstract and conclusion.
- 4. Key words: remove all terms already in title (e.g., slaughterhouse), consider adding One Health as a key word.
RE: terms presented in title were removed and One Health was added to the keywords as suggested.
- 5. Line 46: remove ‘Today’ and start with ‘Food…
RE: removed.
- 6. Line 57: the word ‘safety’ seems to be missing from this sentence. Although, for lines 57-60, do the authors instead mean the process of guaranteeing the origin and type of animal from which an edible item is derived?
RE: the reviewer is right in his/her remark. The word “safety” was added to the phrase.
- 7. Lines 80-81: is there a reference or example of a regulation to cite for hygienic practices and good manufacture practice in this regard? So that readers might have a sense of what these could consist of? Otherwise it is vague.
RE: Yes, there is specific policy. The Regulation (CE) 852/2004. Reference has been added to the text.
- 8. Lines 90-91: what does an ‘all in/all out’ scheme consist of? A brief explanation would provide further context. Also, how do visual only versus classical inspections differ?
RE: A brief explanation of all in/all out has been added to the text.
- 9. The authors allude to the differences, but it would be helpful if they can briefly compare what each consists of. Later in the manuscript I saw this had been more fully developed in the conclusion. The discussion can remain as such there, but more explanation is required for the introduction and background given the importance of these inspections to the rest of the information discussed in this review.
RE: differences between classical meat inspection and visual-only meat inspection, policy have been added to the text
- 10. Lines 111-112: does ‘condemnation’ pertain to the living animal or the derived meat products?
RE: corrected as suggested.
- 11. Line 121: the slaughterhouse represents OR the slaughterhouse offers…
RE: corrected as suggested.
- 12. Lines 128 – 129: change ‘interesting to’ for ‘valuable for’.
RE: corrected as suggested.
- 13. Lines 148 – 149: ‘In addition, few studies…’ seems an odd statement to make given that this is a review paper. Perhaps rephrase to reflect that no or few such studies were encountered during the review?
RE: the phase was corrected as suggested.
- 14. Table 2: is it possible to include the country from which causes of carcass condemnation were reported, similar to table 3? It is more informative and contextualized that way.
RE: countries were added to the table 2 as suggested.
- 15. Lines 173-174: Perhaps use a word other than ‘disgusting’ here – like concerning, distressing, unpalatable, unsavoury.
RE: the word has been replaced by “unsightly”.
- 16. Line 188: clarify that the rate of development in farm animals may be affected by the season/time of year.
RE: explanation was added to the text.
- 17. Lines 189-190: ‘Although the author does not specify the type of health conditions, there is an increased risk of bovine respiratory syndrome’ Is this something the authors are aware of due to their own backgrounds? This should be clarified in relation to what the author of the cited study does not themselves say.
RE: One of the risk factors of respiratory bovine syndrome is the cold season in which factors such as low temperatures, humidity or wind speed increases its prevalence. This is associated to a stress of the immune system of steers increasing the infection susceptibility as reported by Paladino et al., 2021. This information was added to the text.
- 18. Lines 190-192: Why is the food generally of lower quality in the winter?
RE: It is associated to less forage availability and/or adequate volumes of pasture. This information was added to the text.
- 19. Table 3 – I wonder if there might be a different and more graphical way to represent this data? Perhaps on an organ-by-organ basis?
RE: we understand the reviewer´s suggestion. In this review, one of the critics authors made regarding meat inspection is the lack (even absence) of data about carcass and organ condemnations. Since causes of partial condemnations of each organ differs among studies, it is difficult to merge.
- 20. Line 258: Suggest deleting ‘Therefore’ the implication of causality is misleading. Actually, the word ‘Therefore’ is often used somewhat misleadingly throughout the text.
RE: deleted as indicated. Also, the text was revised and improved.
- 21. Line 318: Which disease?
RE: The disease is Mycobacterium avium complex. This information was added to the text.
- 22. Lines 339-341: ‘Although respiratory problems are one of the main health problems in livestock intended for meat production, the lower condemnation rate could be explained by the fact that the pigs are raised in farms with controlled environmental conditions.’ The statement around pigs being raised in farms with controlled environmental conditions seems like a generalization. Is there a reference for this? Or can the authors reword to say this is typically the case?
RE: a reference was added to the text.
- 23. Lines 351-353 – ‘by the year’ is a bit vague, please clarify. Do the authors mean seasonality? Same question for subsequent mentions of ‘year’.
RE: the phase was rewritten as suggested.
- 24. Lines 392-393: ‘Although some welfare indicators have been reported in the literature, their measurement in the slaughterhouse is not even possible.’ Can the authors provide a few examples of such indicators, and explain why it would not be possible to assess them in a slaughterhouse?
RE: some examples and reasons of not be possible to assess in a slaughterhouse have been added to the text.
- 25. Line 393: In accordance with the above comment, change ‘Thus’ to a term like ‘Instead’ or ‘As such…’
RE: corrected as indicated.
- 26. Lines 414-415: cattle beaten more frequently at market? Than other types of animals? Clarify.
Re: since cattle is bigger than small ruminants and pigs, to guide cattle from pasture to the premise throughout paths is carried out by farmers helped by stick that (regrettably), hit cattle to improve its movement.
- 27. Lines 416-417: do the authors also think additional indices or metrics around bruising would be warranted? Presumably, photographic evidence consistent with different types and causes of bruising could be gathered to facilitate inspections? I also wonder about infra-red and heat sensors to distinguish recent from old bruising? I realize this is a little outside the scope of the review, but it could fall under Recommendations.
RE: Control of bruises at slaughterhouse is controversial. Bruises are usually removed during deboning. If a carcass displayed several bruises, the entire carcass can be condemned by alteration of aspect (e. g. repugnant meat). Currently, in EU there is not any official control program at slaughterhouses, so control measures such as photographs (as suggested by reviewer) are out of context. However, discussion about potential control measures was added to the text.
- 28. Line 507: by ‘inappropriate’ use of antibiotics, do the authors mean excessive? Or something else? The term inappropriate is vague.
RE: corrected as suggested.
- 29. Lines 508-510: Exactly how is antibiotic resistance monitored at slaughterhouses?
RE: surveillance of AMR at slaughterhouses is defined by specific policy in the EU under Directive 2003/99/EC and Commission Implementing Decision (EU) 2020/1729 as indicated in references.
- 30. Line 529: replace ‘sensible’ with ‘sensitive’
RE: replaced as indicated.
- 31. Line 554: I am not entirely clear on the meaning of occupational diseases as used by the authors – transferred between farmer and animal?
RE: yes. The term “occupational” refers the infection of farmer, veterinarians, slaughterhouse personnel, etc. in the workplace. To improve the understanding, definition of occupational disease was added in brackets to the text.
- 32. Line 559: Should be One Health.
RE: corrected as indicated.
- 33. Line 562: Their – not Its – prevalence, unless the authors mean either TB or brucellosis.
RE: corrected as suggested.
- 34. Table 8: this is peripheral to my area of expertise, and (full-disclosure) I have dyslexia, but whatever the reason, this table is IMPOSSIBLE for me to read and understand.
RE: we agree the reviewer comment. The table in pdf file is fully unformatted and not fit in page as send to the editorial office.
- 35. Table 9: I suggest re-organizing the data in the prevalence column so that it is presented either from lowest incidence to highest or vice versa in the case of each animal. Otherwise the information seems very chaotic.
RE: we agree with the reviewer´s comment. Prevalence values were re-organized and presented from the highest to the lowest.
- 36. This sentence: ‘With respect to pigs, a higher prevalence of hydatidosis in breeding pigs (0.11%) than in fattening pigs (0.008%) has been observed in Switzerland’ and the associated data does not appear in Table 9 – so are these percentages not from slaughterhouses?
RE: information of prevalence in pig indicated in text was added to the table 9.
- 37. Lines 739-740: What is it about postmortem evaluation that results in scarce data? Somewhere around.
RE: the phrase was removed.
- 38. Line 753, there could be scope for re-purposing some of the information into a section on protective measures consumers can take (e.g., proper cooking).
RE: preventive measures have been added to the test.
- 39. Lines 841-842: a low prevalence in all countries?
RE: Yes. Data was obtained from the EFSA report. Low prevalence of foodborne pathogens across Europe was added to the text.
- 40. Important questions that arose while reading the paper: What exactly happens to condemned carcasses and organs? How are they disposed of? This is highly relevant to scavenging wildlife health and free-roaming scavengers, which in turn has implications for human and livestock health.
RE: disposal by-products of slaughterhouses are not described in text since in Europe, all by product are destroyed by incineration. Classification, traceability, transportation, and destruction are defined by law, specifically by Regulation (EC) No 1069/2009 of the European Parliament and the council of 21 October 2009 laying down health rules as regards animal by-products and derived products not intended for human consumption and repealing Regulation (EC) No 1774/2002 (Animal by-products Regulation. Since in other countries (mainly in developing countries) burial and landfilling is a common practice, it may imply certainly risk for animal and public health. However, authors have not included information about these topics as well other topics such as wastewater management or health risk of slaughter staff. The main objective of the review is highlighting the potential of slaughterhouse as surveillance point of several aspects (e. g. AMR, welfare, etc.) that can improve both animal and human health in cost-effective way.
- 41. Are there any repercussions/sanctions imposed on farmers when bruising is seen? Any animal welfare indices developed around bruising in meat more as a welfare concern? For example, the discussion about considering several tangible conditions together to assess welfare (e.g., hoof condition) was really interesting and useful.
RE: No, there is no sanctions. If carcass presents several bruises, veterinary meat inspector condemned by color alterations or disgusting aspect. Also, no sanctions are applied to farmers in case of hoof diseases.
- 42. Just what does monitoring at slaughterhouse consist of? Clarify a bit more about the different monitoring schemes described (AMR, welfare) – is someone assigned to observe and gather samples? Where do these samples go?
RE: the present review tries to highlight several surveillance activities that can be monitored at slaughterhouse. In the European Union, monitoring of specific zoonotic microorganisms are defined under the Commission Regulation (EC) No 2073/2005 of 15 November 2005 on microbiological criteria for foodstuffs (Text with EEA relevance). Regarding AMR, surveillance of specific resistances are also defined under the Commission Implementing Decision (EU) 2020/1729 of 17 November 2020 on the monitoring and reporting of antimicrobial resistance in zoonotic and commensal bacteria and repealing Implementing Decision 2013/652/EU. However, other aspects discussed in the review such as welfare, monitoring of other zoonotic are not monitored (because they are not compulsory) and nobody is assigned to these.
- 43. The conclusions fall a bit flat after so much work was evidently put into this extensive review.
RE: the conclusion has been revised and improved.
- 43. What about Recommendations and Future Work or Monitoring?
RE: recommendations, monitoring requirements and future work have been added to the conclusion section.
- 44. Define all acronyms in the first instance (OIE, etc.)
RE: acronyms were revised along the text.
- 45. Consider alternating ‘surveillance’ and ‘monitoring’ to reduce repetition.
RE: the text was entirely revised and improved.
- 46. Standardize: farm-to-table OR farm-to-fork.
RE: standardized as suggested.
- 47. I suggest moving information about the different types of inspections (visual, traditional) described in the Conclusions right up to the Introduction.
RE: the text was entirely revised and improved.
- 48. Careful with use of ‘Therefore’ – suggests correlations that do not exist
RE: the text was entirely revised and improved.
- 49. A lot of repetition, ‘also’ ‘in addition’.
RE: the text was entirely revised and improved.
- 50. I would like to see some qualifiers around some of the more absolute statements.
- 50. 1. For example, to the statement (line 570): ‘The prevalence of brucellosis in slaughterhouse varies between 3% and 15% [174-178]’, preface with ‘According to our review, the prevalence…’ Since lower or higher prevalence may also be occurring that has not been captured by the literature
RE: suggestion was considered to improve the text.
- 50. 2. Another example, Lines 812-815: ‘most research is carried out in developing countries where aspects such as implementation of prophylaxis programs, farm biosecurity, animal identification and/or traceability systems and disease eradication programs are scarce or event absent.’
RE: suggestion was considered to improve the text.
REVIEWER 3
- 1. Line 30. Something is wrong with this sentence. Suggest to rewrite.
RE: the sentence was rewritten.
- 2. Line 48. OIE changed names to WOAH. Needs to be changed throughout the publication.
RE: corrected across the text.
- 3. Lines 62-63. Bit odd sentence, reword.
RE: sentence rewritten as indicated.
- 4. Could be nice to include a figure on this with how the different elements relate to each other and play a role in food safety, and the role of the OVI
RE: due to the length of the review, we consider that the addition of a photograph will not increase the understanding of the work.
- 5. Line 86-119 This is a topic on its own, the move from the classic to visual-only inspection. However important, it might be left out as the publication is already extremely long or afterward put as a small chapter in the discussion section.
RE: we totally agree with the reviewer´s comment. In the present review, we refer to the visual-only inspection as a critique of the classic meat inspection. Although some works have indicated that there are no significant differences between classic inspection vs visual inspection in relation to public health, this can only be applied in specific cases (poultry and pig) due to homogeneity (i. e. controlled conditions) in its production, as indicated in the text. However, this cannot be applied to other species (e. g. cattle, small ruminants) and to those small-sized, extensive and/or ecological farms. Until it could be discussed in another section, the authors decided to leave it in the introduction section since the main objective of the review are those points that the slaughterhouse can be used to control and improve animal health and public health.
- 6. Line 146-147: I am not aware that all EU countries have a national database on total cattle condemnation. In my knowledge a few have, most not.
RE: we agree with the reviewer comment. The sentence was rephrased.
- 7. Table 1. Not a percentage.
RE: corrected as indicated
- 8. Suggest to add country as in other Tables.
RE: added as suggested
- 9. To make the tables shorter, I would suggest to combine the studies per country EG. In first part table 3 combine the many studies on liver condemnation causes in Ethiopia. You can put the percentage median and min/max, if there were several studies
RE: we understand the reviewer´s suggestion. In this review, one of the critics authors made regarding meat inspection is the lack (even absence) of data about carcass and organ condemnations. Since causes of partial condemnations of each organ differs among studies, it is difficult to merge.
- 10. Species - typo on several places.
RE: corrected as indicated.
- 11. Define S&G sheep and goats.
RE: corrected as indicated.
- 12. A much discussed indicator to measure pig welfare is tail docking/tail biting recording. The EC currently has even a call to fund a project developing automatical monitoring tolls.
RE: Information about the project was added to the text as suggested.
- 13. Unfortunately, all use, not only inappropriate or continued use, can lead to resistance.
RE: the phase was modified as suggested.
- 14. The conclusion has quite a bit of repeat of things said above. Would make it shorter.
RE: the conclusion was revised and improved.
- 15. Something wrong with sentence. Maybe break up in two sentences.
RE: the sentence was rewritten.
Reviewer 2 Report
Comments attached.
Additionally, it would be very interesting and relevant to know more about any reported practices around disposal of, or interaction with, condemned carcasses and organs. This has huge relevance to human, livestock and wildlife health too.

Author Response

(The authors gave the same response as above.)

Reviewer 3 Report
Congratulations on this very extensive study on how slaughterhouse data can be better used for animal and public health reasons. This systematic review brings together data from many studies all over the world and gives a very detailed overview.
My main problem due to the length of the papers and the enormous amount of references quoted, is that people get lost in the results. I would suggest to try to shorten the text a bit, shorten the tables (e.g. by combining studies from the same country) and highlighting better the main outcomes per chapter (e.g. on animal welfare, on AMR monitoring, etc). This would increase enormously the readability of this excellent publication.
I further made more detailed comments in the file attached.

Author Response

(The authors gave the same response as above.)

Round 2
Reviewer 2 Report
I want to commend the authors because I think this review is very, very interesting and offers an important contribution. I also greatly appreciated their attentiveness and openness to the reviewers' edits and comments, including mine. The reorganization of the tables has made them clearer and more informative!
Regarding my recommendation that the written English be extensively edited (by the journal) to standardize the final manuscript, I hope the authors understand this is not a criticism, and that they have done really well writing such a technical paper in a secondary language.
I am very satisfied with the way my comments and concerns have been incorporated into this reworked version, thank you for that. The only outstanding concern that I have regards the disposal of condemned organs and carcasses. I had said:
Important questions that arose while reading the paper: What exactly happens to condemned carcasses and organs? How are they disposed of? This is highly relevant to scavenging wildlife health and free-roaming scavengers, which in turn has implications for human and livestock health.
And they responded:
Disposal by-products of slaughterhouses are not described in text since in Europe, all by product are destroyed by incineration. Classification, traceability, transportation, and destruction are defined by law, specifically by Regulation (EC) No 1069/2009 of the European Parliament and the council of 21 October 2009 laying down health rules as regards animal by-products and derived products not intended for human consumption and repealing Regulation (EC) No 1774/2002 (Animal by-products Regulation. Since in other countries (mainly in developing countries) burial and landfilling is a common practice, it may imply certainly risk for animal and public health. However, authors have not included information about these topics as well other topics such as wastewater management or health risk of slaughter staff. The main objective of the review is highlighting the potential of slaughterhouse as surveillance point of several aspects (e. g. AMR, welfare, etc.) that can improve both animal and human health in cost-effective way.
For transparency and to add even further value to this review, I still think it would be worthy and important to include this information, perhaps as a footnote or, better, in the Introduction. I would argue that even if in Europe the procedure is for all by-products to be destroyed by incineration, this is not necessarily common knowledge or something people would know or assume. Therefore there is a chance here to raise awareness for the reader that these peripheral concerns and mechanisms do in fact exist for disposal of condemned organs and carcasses. In One Health the disposition of tainted carcasses and biproducts is an important consideration and I think this is an excellent opportunity just to impart that extra bit of detail (what they have responded above) which might otherwise be overlooked.
Author Response
REVIEWER´S 2 COMMENTS
- I want to commend the authors because I think this review is very, very interesting and offers an important contribution. I also greatly appreciated their attentiveness and openness to the reviewers' edits and comments, including mine. The reorganization of the tables has made them clearer and more informative!
Regarding my recommendation that the written English be extensively edited (by the journal) to standardize the final manuscript, I hope the authors understand this is not a criticism, and that they have done really well writing such a technical paper in a secondary language.
I am very satisfied with the way my comments and concerns have been incorporated into this reworked version, thank you for that.
RE: the authors are grateful for the exhaustive revision carried out by the reviewer that has served to improve the work. We also thank the reviewer for congratulations and the interest for the review paper done. We inform the reviewer the entire manuscript was revised and improved.
- The only outstanding concern that I have regards the disposal of condemned organs and carcasses. For transparency and to add even further value to this review, I still think it would be worthy and important to include this information, perhaps as a footnote or, better, in the Introduction. I would argue that even if in Europe the procedure is for all by-products to be destroyed by incineration, this is not necessarily common knowledge or something people would know or assume. Therefore there is a chance here to raise awareness for the reader that these peripheral concerns and mechanisms do in fact exist for disposal of condemned organs and carcasses. In One Health the disposition of tainted carcasses and biproducts is an important consideration and I think this is an excellent opportunity just to impart that extra bit of detail (what they have responded above) which might otherwise be overlooked.
RE: information about total and/or partial (organs) condemnations was added in the introduction section.